# PuzzleFusion++: Auto-agglomerative 3D Fracture Assembly by Denoise and Verify

**Zhengqing Wang**[1,*] **Jiacheng Chen**[1,*] **Yasutaka Furukawa**[1,2]

[1] Simon Fraser University, [2] Wayve

`{zwa170,jca348,furukawa}@sfu.ca`

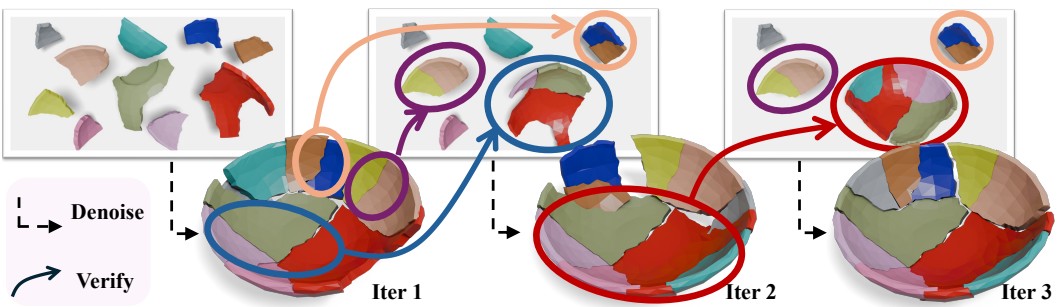

Figure 1: PuzzleFusion++ iteratively aligns and assembles fracture fragments into a 3D shape, resembling how humans solve jigsaw puzzles. At each iteration, a diffusion model solves for the 6-DoF alignments of the fragments, and a transformer verifies the pairwise alignments and merge them into larger fragments. We call our approach "auto-agglomerative," referring to auto-regressive methods for the iterative process and agglomeration clustering for the hierarchical grouping.

## Abstract

This paper proposes a novel "auto-agglomerative" 3D fracture assembly method, PuzzleFusion++, resembling how humans solve challenging spatial puzzles. Starting from individual fragments, the approach 1) aligns and merges fragments into larger groups akin to agglomerative clustering and 2) repeats the process iteratively in completing the assembly akin to auto-regressive methods. Concretely, a diffusion model denoises the 6-DoF alignment parameters of the fragments simultaneously, and a transformer model verifies and merges pairwise alignments into larger ones, whose process repeats iteratively. Extensive experiments on the Breaking Bad dataset show that PuzzleFusion++ outperforms all other state-of-the-art techniques by significant margins across all metrics In particular by over 10% in part accuracy and 50% in Chamfer distance. The code and models are available on our project page: https://puzzlefusion-plusplus.github.io.

## 1 Introduction

Humans have an innate proficiency in solving complex spatial puzzles. Starting with scattered pieces, we assess connections based on their shapes and fits, gradually merge them into larger components, and repeat the process through trial and error until forming a single assembly. The task requires a combination of sophisticated skills on shape analysis, spatial reasoning, and trial and error process of navigating the massive combinatorial solution space.

A computational system with such capabilities would have significant impacts across broad domains. For example, archaeologists could reassemble fragments and uncover ancient artifacts. Forensic specialists could reconstruct broken objects from accident scenes to deduce causative events. Bio-chemistry scientists could discover effective drugs by analyzing compatible protein 3D structures.

---

[*]Equal contribution

Deep neural networks (DNNs) have brought dramatic progress in developing such an intelligent computational method. DNN-based encoder enables robust shape matching by encoding raw 3D scans (i.e., point clouds) into latent embeddings, where a combinatorial optimization technique searches for an optimal set of pairwise alignments (Lu et al., 2023). PuzzleFusion (Hossieni et al., 2023) proposed an innovative fully neural system based on Diffusion Models that simultaneously aligns all the pieces in solving 2D jigsaw puzzles. This paper pushes the frontier of spatial puzzle solving by proposing a fully neural system that simulates how humans tackle this problem.

Starting from individual fracture fragments, the approach 1) simultaneously aligns and merges fragments into larger groups, akin to agglomerative clustering, and 2) repeats this process iteratively in completing the assembly akin to auto-regressive methods. We call our approach "auto-agglomerative", mimicking the way humans tackle challenging spatial puzzles. Concretely, after using PointNet++ (Qi et al., 2017) and VQVAE (Van Den Oord et al., 2017) to encode each fragment into latent embeddings, the approach uses a diffusion model to solve for the 6-DoF alignments of all the fragments. A transformer model verifies the inferred pairwise alignments and merges verified pairs into larger groups. The process repeats until forming a single assembly.

Extensive experiments on the Breaking Bad dataset (Sellán et al., 2022) show that PuzzleFusion++ outperforms all existing methods by significant margins in all metrics, specifically by over 10% in part accuracy and over 50% in the Chamfer distance metric, pushing the frontier of computational methods for challenging spatial puzzle tasks. The code and models are available on our project page: https://puzzlefusion-plusplus.github.io.

## 2 RELATED WORK

**Diffusion models.** Diffusion models (Ho et al., 2020; Song et al., 2020; Sohl-Dickstein et al., 2015) have gained prominence for their exceptional performance in generating images (Ramesh et al., 2022; Rombach et al., 2022; Dhariwal & Nichol, 2021b), videos (Blattmann et al., 2023a;b), and 3D assets (Zeng et al., 2022; Luo & Hu, 2021; Poole et al., 2022; Tang et al., 2024; Alliegro et al., 2023). The great capacity to capture complex data distribution makes diffusion-based models suitable for many tasks beyond generation. Recent works have extended diffusion models as general solvers for deterministic tasks, including object detection (Chen et al., 2023b), camera pose estimation (Wang et al., 2023), shape reconstruction (Cheng et al., 2023; Chen et al., 2023a), and puzzle solving (Hossieni et al., 2023; Scarpellini et al., 2024). Our method exploits diffusion models within an agglomerative puzzle-solving framework to deal with the complex geometric reasoning for the fracture assembly tasks.

**Auto-regressive methods.** Auto-regressive methods recursively predict the next values in a sequence based on the previously observed or predicted elements. Auto-regressive approaches like GPT (Radford et al., 2018) and its follow-ups have revolutionized natural language processing (NLP). Similar approaches also generate images as pixel sequences (Van Den Oord et al., 2016b), audio (Van Den Oord et al., 2016a), or 3D shapes (Nash et al., 2020; Siddiqui et al., 2023), etc. The auto-regressive agglomeration process in our method draws inspiration from these successes. By iteratively aligning and merging fractured fragments into larger ones for future iterations, we gradually assemble all fragments into a single object. The process mimics human cognitive puzzle-solving strategies, aiming for more precise and efficient assembly.

**3D shape assembly.** 3D fracture assembly has been a challenge for machine intelligence (Huang et al., 2006). PartNet (Mo et al., 2019) presents a large 3D object segmentation database (*e.g.*, the legs or seat of a chair), fueling research on learning-based methods (Yin et al., 2020; Li et al., 2023; Narayan et al., 2022). Recent transformer-based approaches (Xu et al., 2024; Du et al., 2024; Zhang et al., 2022) capture the global semantic and geometric contexts well to enhance the alignment accuracy. Chen et al. (2022) proposes a shape-mating dataset where objects are randomly fractured into two fragments, and an adversarial learning framework reassembles the two fragments. Lamb et al. (2023) collects a dataset with real-world broken objects. The Breaking Bad dataset (Sellán et al., 2022) introduces a more challenging benchmark, where diverse objects are fractured into multiple (*i.e.*, 2 to 100) fragments via physical simulations. Jigsaw (Lu et al., 2023) utilizes deep networks for fracture surface segmentation and matching, followed by a global alignment step to derive the poses. SE(3)-equiv (Wu et al., 2023) extracts global equivariant and invariant features to represent each fractured fragment and directly regress the 6-DoF alignment parameters. DiffAssemble (Scarpellini

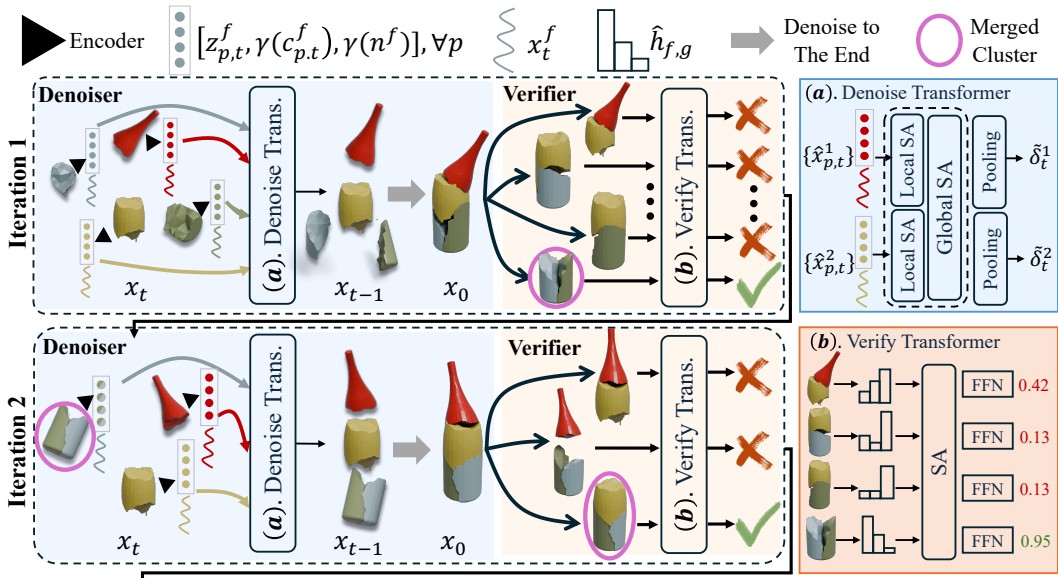

Figure 2: The architecture overview of PuzzleFusion++ (mesh used only for visualization). **Left:** An illustration of the auto-agglomerative fracture assembly process with the first two iterations. **Right**: Close-ups of the SE3 denoise transformer and the pairwise alignment verifier transformer. Please refer to Figure 3 for the details of the architectures.

et al., 2024) further applies a diffusion model to this framework. This paper introduces an auto-agglomerative framework to mimic how humans solve spatial puzzles, where a diffusion model with local geometric features serves as a spatial solver in each iteration, and a transformer verifies the correctness of aligned fragments and merges them into larger groups.

# 3 PuzzleFusion++: An auto-agglomerative approach for fracture assembly

PuzzleFusion++ innovates computational methods for fracture assembly via an "auto-agglomerative" approach, where the task is to take a set of rigid fractured fragments as input and estimate their 6-DoF alignments. In our case, an input fragment and an output alignment are a point-cloud and a 3D rigid transformation, respectively. PuzzleFusion++ consists of two modules, the denoiser and the verifier. The denoiser aligns individual fragments (§3.2), where the verifier classifies the correctness of pairwise alignments and merges verified pairs into larger fragments (§3.3), akin to agglomerative clustering. The approach repeats the process based on previous predictions until forming a single 3D shape, akin to auto-regressive methods (§3.4).

## 3.1 Preliminaries

**Input representation.** An input fragment comes as point clouds. We train PointNet++ (Qi et al., 2017) and VQ-VAE (Van Den Oord et al., 2017) to encode the point clouds into latent vectors at the points (details in Section A.1). PointNet++ employs furthest point sampling to select 25 points, creating 25 corresponding point latent vectors for each fragment. This approach retains intricate surface details, offering an advantage over the global feature extraction used in previous studies (Wu et al., 2023; Scarpellini et al., 2024). Note that the encoding process normalizes the 3D coordinate frame so that the center of mass is at the origin, and the longest dimension of the axis-aligned bounding box becomes a unit length. The encoding is still sensitive to rotation, where latent vectors are re-calculated as we optimize fragment alignments.

**Output representation.** The output alignment is represented as a 7D vector for each fragment, composed of a 4D quaternion and a 3D translation vector.

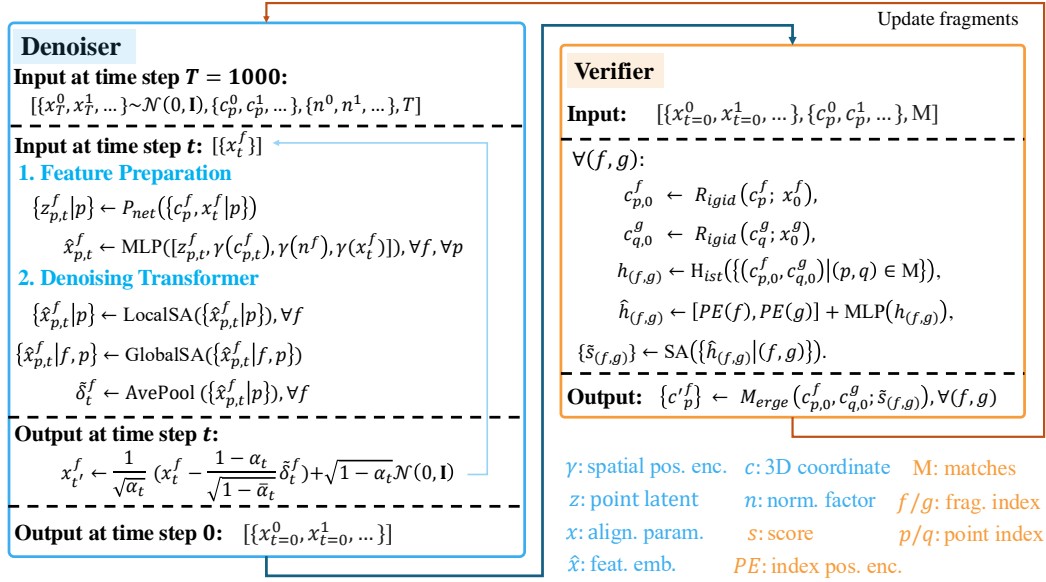

Figure 3: Inference pipeline with architecture specifications. The denoiser (blue) is a diffusion model with Transformer architecture at its core. The verifier (orange) is a Transformer.

**Anchor fragments.** Fracture assembly has an inherent ambiguity in the 3D rigid transformation. We introduce the concept of *anchor fragments* to settle down a reference coordinate system. The anchor fragments have fixed alignments (*i.e.*, the identity matrix for rotation and the zero vector for translation) and do not move in the assembly process. At the beginning of the inference, the largest fragment (based on the longest dimension of its axis-aligned bounding box) becomes an anchor. In the assembly process, any fragment that merges with an anchor also becomes an anchor. During training, we set anchors to be the largest fragment (with the same definition as above) plus its neighboring fragments with a 50% chance each to simulate the inference-time merging process.

## 3.2 SE3 DENOISER

**Forward process.** Let $x_t^f$ denote the 7D alignment vector of fragment $f$ at timestep $t$. $x_0^f$ is the ground-truth alignment before noise injection. A piece-wise quadratic scheduler adds a Gaussian noise (except to the anchor fragment) $\delta_t^f$ to $x_0^f$, using $m$=700 and $T$=1000:

$$x_t^f = \sqrt{\bar{\alpha}_t}x_0^f + \sqrt{1-\bar{\alpha}_t}\delta_t^f, \quad \bar{\alpha} = \begin{cases} 1 - (0.1)\left(\frac{t}{m}\right)^2 & \text{if } t \leq m \\ 0.9\left(1 - \left(\frac{t-m}{T-m}\right)^2\right) & \text{if } m < t \leq T \end{cases} \quad (1)$$

**Reverse process.** Following PuzzleFusion (Hossieni et al., 2023), each sampled point of each fragment keeps track of the alignment estimate in the denoise transformer architecture. The denoising architecture is standard (See Figures 2 and 3), where the main components are 1) Intra-fragment self-attention (Local SA) among 25 sampled points within a fragment; 2) Inter-fragment self-attention (Global SA) among all sampled points across all fragments; and 3) Average pooling over the 25 sampled points of fragment $f$ to predict the residual $\tilde{\delta}_t^f$.

Both intra-fragment and inter-fragment SA modules utilize adaptive layer norm (Dhariwal & Nichol, 2021a) to inject the timestep $t$. Each module comprises 6 self-attention layers, each with 8 multi-heads and a feature dimension of 512.

A feature embedding $\hat{x}_{p,t}^f \in \mathbb{R}^{512}$ of a sampled point $p$ of a fragment $f$ is the concatenation of the four features: 1) The alignment estimate $x_t^f \in \mathbb{R}^7$ with a positional encoding ($\gamma() \in \mathbb{R}^{147}$; 2) The point latent $z_{p,t}^f \in \mathbb{R}^{64}$; 3) The 3D coordinate $c_{p,t}^f \in \mathbb{R}^3$ with a positional encoding ($\gamma() \in \mathbb{R}^{63}$); and

4) The scale normalization factor $n^f$ of the point-cloud (§3.1) with a positional encoding ($\gamma() \in \mathbb{R}^{21}$). MLP maps the concatenated feature into a 512D latent embedding.

Note that anchor fragments are processed exactly the same way, while their alignment should be the identity rotation and zero-vector translation. Therefore, alignment estimation is discarded at every denoising step during inference, and no gradients are injected during training for anchor fragments.

The standard DDPM (Ho et al., 2020) loss is imposed on non-anchor fragments: $E_{t,x_0^f,\delta_t^f} \left\| \delta_t^f - \tilde{\delta}_t^f \right\|^2$.

DiffAssemble (Scarpellini et al., 2024) applies diffusion models to solve both 2D and 3D reassembly problems but obtains suboptimal performance in the complex 3D fracture assembly task. Our denoiser provides tailored designs over all critical components of diffusion models. Please refer to §4.3 for detailed analyses and experimental results.

### 3.3 PAIRWISE ALIGNMENT VERIFIER

Given the alignment of $F$ fragments, the verifier employs a Transformer architecture with $\binom{F}{2}$ nodes to perform binary classification on the correctness of $\binom{F}{2}$ pairwise alignments simultaneously. Note that verifying the correctness of a given alignment is a lot easier than searching for an optimal alignment of many fragments, thus a straightforward Transformer architecture with the following node embeddings suffices for this task.

An input node embedding for a pair of fragments ($f$ and $g$) is the sum of two vectors. The first vector is the concatenation of the sinusoidal positional encodings of the fragment indices, $[PE(f), PE(g)] \in \mathbb{R}^{256}$. The second vector represents the alignment quality of $f$ and $g$, utilizing a point matching module from Jigsaw by Lu et al. (2023). Specifically, the module takes the point clouds of all the fragments (without alignment) and identifies a set of matching points. For point matches between $f$ and $g$, we calculate the Euclidean distances of points and construct a normalized histogram with six bins. We append the total number of matches as the seventh value, which is fed into an MLP to construct the second vector. The distance thresholds of these bins are set at (0, 1e-3, 5e-3, 1e-2, 5e-2, 1e-1, $\infty$). The verifier is optimized using the standard binary cross-entropy loss.

### 3.4 AUTO-AGGLOMERATIVE INFERENCE

The denoiser and the verifier iteratively align and merge fracture fragments into larger groups. This process repeats six iterations or until all the fragments are merged into a single component. Fragment pairs are verified for merging if their classification score exceeds a threshold of 0.9. One challenge in fragment merging is the potential inclusion of points from inner surfaces when taking the union of the point-clouds. We employ simple heuristics to detect and remove such inner points before merging. Specifically, we compute a surface normal for each point of a point-cloud using *estimate_pointcloud_normals* in the *pytorch3d.ops* module. A point is identified as an inner surface point if another point in the opposing point-cloud is within a distance of 0.001 and their surface normals yield a negative dot product. A point-cloud of a fragment contains 1,000 points. After removing inner surface points, we resample to maintain 1,000 points using the farthest point sampling method. For anchor fragments, we preserve them as separate and freeze their alignments rather than merging to retain the high-resolution information crucial for large (e.g., anchor) fragments.

## 4 EXPERIMENTS

We use a server with four NVIDIA RTX A6000 GPUs for experiments. The denoiser is trained for 2000 epochs with a batch size of 64. The initial learning rate is 2e-4 and decays by a factor of 10 at 1200 and 1700 epochs. The AdamW optimizer is used with a decay factor of 1e-6. The verifier is trained for 100 epochs using the same training settings as the denoiser. The training of the denoiser and the verifier takes approximately 75 hours and 10 minutes.

**Datasets.** Following recent works (Wu et al., 2023; Lu et al., 2023), we use the Breaking Bad dataset (Sellán et al., 2022). Specifically, 34,075 assemblies from 407 objects in the *everyday* subset are for training. 7,679 assemblies from 91 objects in the *everyday* subset and 3,651 assemblies from 40 uncategorized objects in the *artifact* subset are for testing.

Table 1: Quantitative evaluations on the Breaking Bad dataset. The numbers for the five baselines (marked as *) are copied from their papers, except for the CD metric of Jigsaw which we calculated by running their official codebase. SE(3)-Equiv reported numbers for assemblies with at most 8 fragments in their paper, where we modified their codebase to calculate the metrics under our setting (i.e., for assemblies with at most 20 fragments). The running speed is measured on a single RTX NVIDIA 4090 GPU by running the official codebase if available. The cyan and the orange texts denote the best and the second best results, respectively.

| Method | RMSE(R) ↓ degree | RMSE(T) ↓ $\times 10^{-2}$ | PA ↑ % | CD ↓ $\times 10^{-3}$ | Speed ms/sample |
|---|---|---|---|---|---|
| _Trained and tested on the_ `everyday` _subset_ | | | | | |
| Global (Li et al., 2020)* | 80.7 | 15.1 | 24.6 | 14.6 | 23.7 |
| LSTM (Wu et al., 2020)* | 84.2 | 16.2 | 22.7 | 15.8 | 29.9 |
| DGL (Zhan et al., 2020)* | 79.4 | 15.0 | 31.0 | 14.3 | 28.7 |
| SE(3)-Equiv(Wu et al., 2023) | 79.3 | 16.9 | 8.41 | 28.5 | 129.9 |
| DiffAsb(Scarpellini et al., 2024)* | 73.3 | 14.8 | 27.5 | - | - |
| Jigsaw(Lu et al., 2023)* | 42.3 | 10.7 | 57.3 | 13.3 | 1063.5 |
| PuzzleFusion++ (Ours) | 38.1 | 8.04 | 70.6 | 6.03 | 928.5 |
| _Trained on the_ `everyday` _subset, tested on the_ `artifact` _subset_ | | | | | |
| Jigsaw(Lu et al., 2023)* | 52.4 | 22.2 | 45.6 | 14.3 | - |
| PuzzleFusion++ (Ours) | 52.1 | 13.9 | 49.6 | 14.5 | - |

We limit assemblies with at most 20 fragments and do not use the assembly category labels (e.g., bottles, vases, and mirrors). Each fragment is represented as a point cloud consisting of 1,000 points, following the baseline settings. Note that the training/testing split is exactly the same as Breaking Bad dataset (Sellán et al., 2022) for fair comparisons.

**Evaluation metrics.** The Breaking Bad dataset offers three evaluation metrics, where the biggest fragment is used to align the predicted assembly with the ground truth before calculating the metrics, which is the same as in Jigsaw (Lu et al., 2023) evaluation settings:

• *Root mean square error (RMSE)* of both the rotation and the translation parameters.
• *Part accuracy (PA)* is the ratio of fragments whose per-fragment Chamfer distance is less than 0.01.
• *Chamfer distance (CD)* is calculated per assembly.

**Pre-processing.** At test time, we apply random rotations to all the fragments and move the center-of-mass of each fragment to the coordinate frame origin to hide all the ground-truth information. At training time, we prepare ground-truth assemblies and their alignment parameters by applying a random rotation to each whole assembly and moving the center-of-mass of its anchor fragment to the coordinate frame origin.

**Competing methods.** We include three baselines from the Breaking Bad dataset and three other existing approaches designed for the fracture assembly task:

• *Global, LSTM, and DGL* are the initial baselines provided by the Breaking Bad dataset. Global combines the global shape features with per-fragment features and directly regresses the pose. LSTM focuses on learning cross-fragment relationships, using a bi-directional LSTM to predict pose. DGL utilizes the graph neural network to find the relationship between fragments. These methods are originally designed for PartNet (Mo et al., 2019) assembly tasks.

• *SE(3)-Equiv* (Wu et al., 2023) integrates equivariant and invariant features to model multi-part correlations. The released code only trains and tests on samples with at most 8 fragments. We train and test their system for assemblies with at most 20 fragments for fair comparison.

• *DiffAssemble* (Scarpellini et al., 2024) uses the equivariant encoder to extract per-fragment features similar to SE(3) (Wu et al., 2023) and uses a diffusion model to predict the pose.

• *Jigsaw* (Lu et al., 2023) is the state-of-the-art fracture assembly method, leveraging hierarchical features of global and local geometry to do fracture surface point matching and recover the poses.

## 4.1 QUANTITATIVE EVALUATIONS

Table 1 demonstrates that PuzzleFusion++ outperforms all the six baselines across all the metrics with significant margins, except one metric in one case to be discussed below. Among the six baselines, Jigsaw is a clear winner that boasts a learning-based point matcher, leveraging global and local geometry. However, Jigsaw is not a fully neural system, relying on classical optimization in searching for a compatible set of point matches. PuzzleFusion++ is fully neural, directly searches for a compatible alignment of fragments with a powerful diffusion model, and repeats the process many times as confident partial alignments are merged into bigger fragments, which is precisely how humans would solve the task.

The lower section of Table 1 evaluates the generalization capabilities of Jigsaw and PuzzleFusion++. Both models are trained on the *everyday* subset and tested on the *artifact* subset. Jigsaw marginally surpasses PuzzleFusion++ in the CD metric (14.5 vs 14.3), where PuzzleFusion++ suffers from a major performance decline across all metrics compared to Jigsaw. Despite apparent superior generalization by Jigsaw, we argue that this is a side-effect of powerful global geometry learning by PuzzleFusion++, which learns what "everyday" objects are at training and fail on "artifact" objects at test time. Jigsaw focuses more on local geometry learning which is less affected by the change of an object category.

Table 2: Ablation study on the number of iterations.

| Method | RMSE (Rot.) | RMSE (Trans.) | PA | CD | Speed (ms) |
|---|---|---|---|---|---|
| Jigsaw | 42.3 | 10.7 | 57.3 | 13.3 | 1063.5 |
| Ours (#$ite$=1) | 40.8 | 9.06 | 67.3 | 6.45 | 243.5 |
| Ours (#$ite$=2) | 39.4 | 8.48 | 68.8 | 6.28 | 429.3 |
| Ours (#$ite$=4) | 39.1 | 8.23 | 69.8 | 6.15 | 685.2 |
| Ours (#$ite$=6) | 38.1 | 8.04 | 70.6 | 6.02 | 928.5 |

Table 3: Ablation study on the number of sampling steps when only using the denoiser.

| Steps | RMSE (Rot.) | RMSE (Trans.) | PA | CD | Speed (ms) |
|---|---|---|---|---|---|
| 5 | 46.1 | 10.3 | 62.4 | 7.79 | 76.8 |
| 10 | 42.2 | 9.26 | 66.4 | 6.65 | 132.6 |
| 20 | 40.8 | 9.06 | 67.3 | 6.45 | 243.5 |
| 50 | 40.9 | 9.09 | 67.2 | 6.86 | 589.1 |

Table 4: Ablation study on the core design of the denoiser.

| Scheduler | Autoencoder | Anchor | RMSE (Rot.) | RMSE (Trans.) | PA |
|---|---|---|---|---|---|
| × | ✓ | ✓ | 48.2 | 11.9 | 57.5 |
| ✓ | × | ✓ | 53.5 | 10.5 | 56.9 |
| ✓ | ✓ | × | 47.4 | 10.7 | 61.1 |
| ✓ | ✓ | ✓ | 40.8 | 9.06 | 67.3 |

Table 5: Ablation study on different types of verifier.

| Method | Verifier Type | RMSE (Rot.) | RMSE (Trans.) | PA |
|---|---|---|---|---|
| Ours (#$ite$=1) | None | 40.8 | 9.06 | 67.3 |
| Ours (#$ite$=6) | Jigsaw | 38.1 | 8.04 | 70.6 |
| Ours (#$ite$=6) | GT (Upper Bound) | 34.0 | 5.87 | 82.9 |

Table 6: Ablation on verifier's performance

| Threshold | Accuracy (%) | Precision (%) | Recall (%) |
|---|---|---|---|
| 0.5 | 87.99 | 58.64 | 73.51 |
| 0.9 | 90.77 | 87.88 | 45.95 |

Table 7: Random initializations of the anchor fragment.

| RMSE(R) | RMSE(T) | PA | CD |
|---|---|---|---|
| $40.86 \pm 0.37$ | $9.03 \pm 0.18$ | $67.49 \pm 0.51$ | $6.69 \pm 0.62$ |

## 4.2 QUALITATIVE EVALUATIONS

Figure 4 visualizes the assembly results of PuzzleFusion++ and baselines for representative examples. The §B provides many more results, organized by the number of fragments in each assembly without cherry-picking to provide a broad and unbiased selection. For each result, we put the number of correctly assembled fragments and the number of total fragments. We show two easy examples with less than 6 fragments in the first two rows, where Jigsaw obtains similar results to ours. The rest of the figure presents challenging examples with more than 10 fragments. As the number of fragments increases, all baselines have trouble making reasonable assembly while PuzzleFusion++ produces much more accurate and reasonable results. Specifically, the potential ambiguity in local geometric features could deteriorate the performance of Jigsaw when there are too many small fragments, while our approach stays robust by exploiting high-level geometric reasoning in the auto-agglomerative

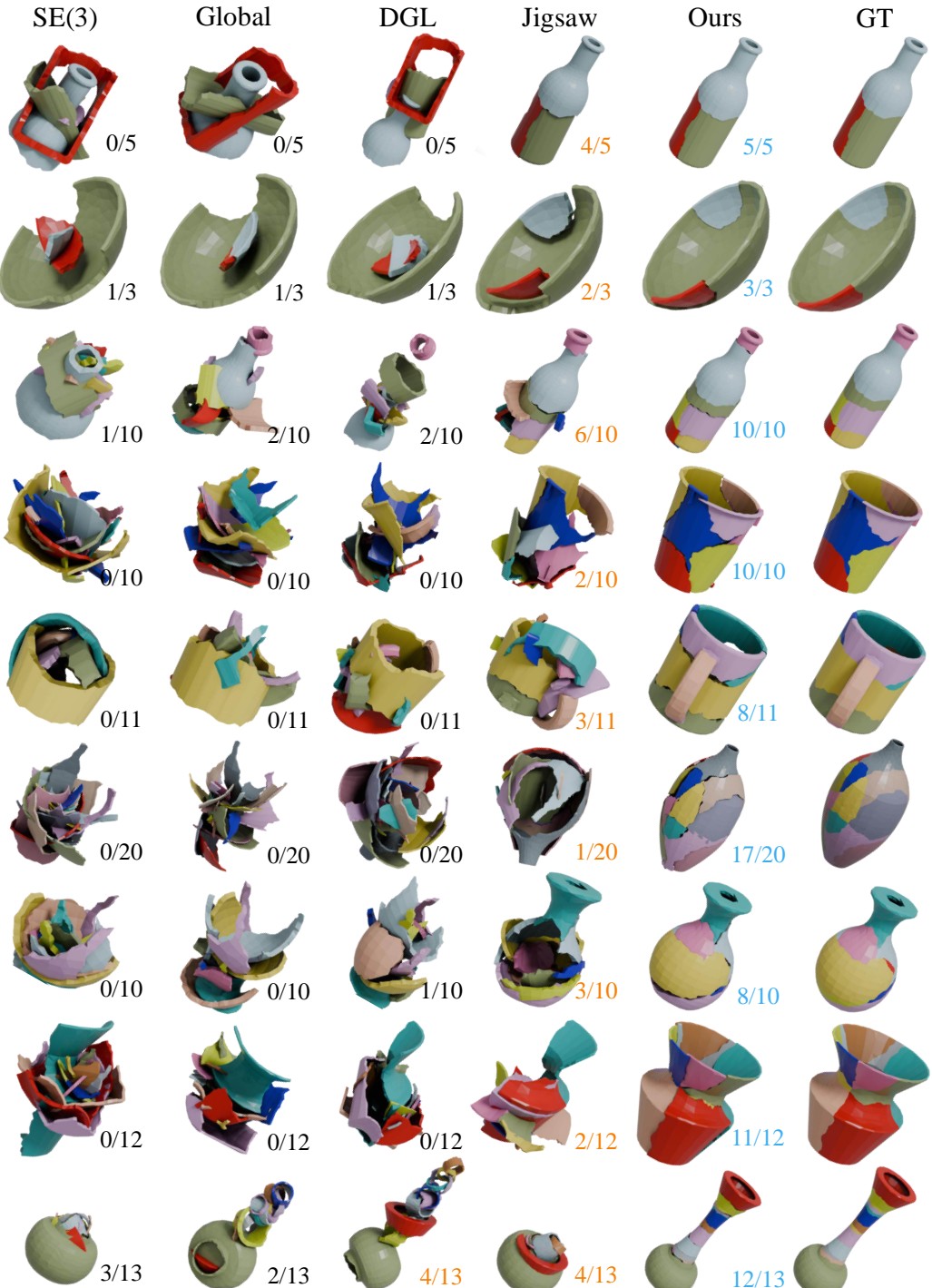

Figure 4: Qualitative comparisons on the Breaking Bad dataset (mesh used only for visualization). We transform the assembled objects to same coordinate system and normalize their sizes for clearer visualization. For each result, we show the number of successfully assembled fragments versus the total number of fragments (*i.e.*, the part accuracy metric). Please see the Appendix for additional results.

process. Please refer to §B for direct comparisons between Jigsaw and our PuzzleFusion++, and the supplementary video for the animations.

### 4.3  ABLATION STUDIES AND ANALYSES

**Iterations of the auto-agglomerative process.** Table 2 presents the ablation study on the number of auto-agglomerative iterations (#*ite*): The second row (#*ite*=1) demonstrates that our denoiser alone performs better than the other baselines (Table 1).

.

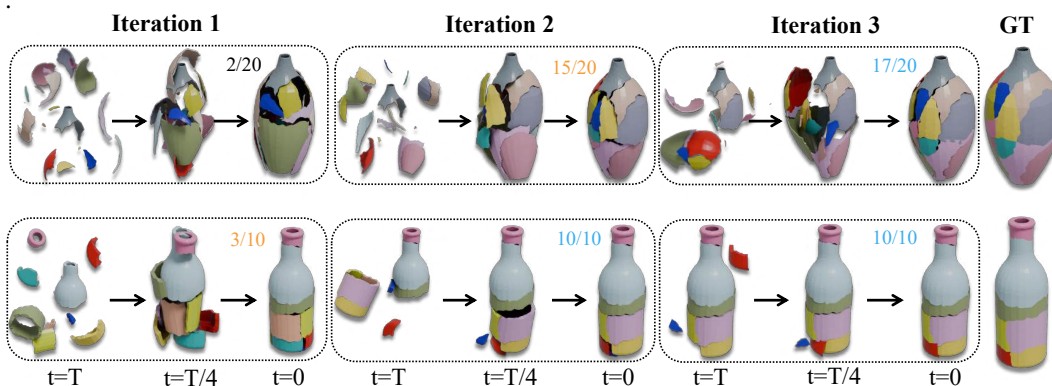

Figure 5: Visualization of the assembly process in the first three auto-agglomerative iterations (mesh used only for visualization). We show two challenging examples with more than 9 fragments. The number of successfully assembled fragments increases as the system runs for more iterations.

The rest of the table shows that our performance consistently improves with more auto-agglomerative iterations (*i.e.*, iterating the denoiser and the verifier). The iterations are particularly important for complex assemblies with many fragments, as shown by two hard examples in Figure 10. The verifier finds a few small fragments correctly aligned in the early iterations, which simplifies the problem for the denoiser and helps complete the assembly in the later iterations.

**Denoiser.** Table 3 shows how performance varies with different numbers of sampling steps when only the denoiser is used during testing. The optimal performance is achieved with 20 sampling steps, yet performance remains robust even with fewer steps.

DiffAssemble (Scarpellini et al., 2024) was initially designed to solve 2D jigsaw puzzles using a diffusion model, and it directly extends the 2D architecture to tackle the 3D variant without proper designs for 3D. While achieving SOTA performance on the 2D jigsaw puzzle, it fails to produce reasonable results on the 3D task – our denoiser-only result outperforms it by a huge margin (67.3% vs. 27.5%). Table 4 shows the ablation of our key denoiser designs: 1) pre-training the autoencoder to learn better local geometric features for fractures, 2) introducing an anchor fragment to resolve the ambiguity in 3D assembly, and 3) adjusting the sampling scheduler to balance the rough and fine alignments. These three changes bring performance closer to the baseline method, Jigsaw. Additionally, our research contribution (auto-agglomerative formulation) provides a significant advantage over Jigsaw in addition to these improvements.

**Verifier.** Table 6 shows the performance of the verifier with different thresholds. In practice, we use the threshold of 0.9 for the transformer's output to be classified as correct alignment. This high threshold ensures that the selected pairs have high confidence, which is crucial since our framework does not backtrack errors. The verifier's accuracy is 90.77%, which is high because all pairs are input into the transformer, and most pairs are trivially not matched (*i.e.*, they do not have any matching points). The lower recall (45.95%) indicates the model misses many true matches.

The suboptimal accuracy of the verifier limits the gain brought by increasing the number of auto-agglomerative iterations. To understand the upper bound of our method, we experiment with a "ground-truth verifier", which always makes the correct merging of pieces. Note that this verifier does not leak ground-truth pose information to the denoiser but rather provides correct high-level piece-

merging guidance. Interestingly, Table 5 shows that this leads to significantly better performance than the verifier using Jigsaw matchings. A more robust verifier model, as well as a potential backtracking strategy, are promising directions for future work.

**Random initialization of the anchor fragment.** Table 7 presents the results of 10 model inferences with different random initializations of the anchor fragment and calculates the mean and standard deviation of the metrics. Different anchor fragment initialization does not affect the quality of the assembly results. In addition, Figure 6 shows one example assembly process with different anchor initialization. The anchor fragment is randomly rotated and moved to the origin during both training and testing. As explained in §3.1, it helps set up a fixed reference coordinate frame for the assembly process, resolving the inherent ambiguity in 3D rigid transformation.

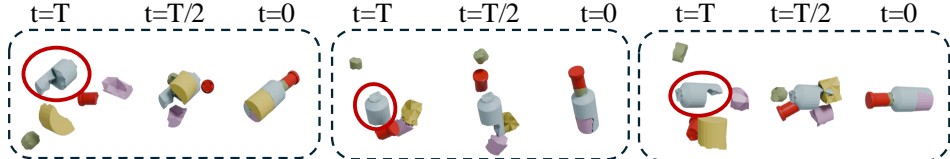

Figure 6: Assembled results of a single example with different anchor initializations from timestep T to 0. The red circle marks the anchor fragment, with the object assembled around it.

### 4.4 FAILURE CASES AND LIMITATIONS

Figure 7 shows two failure modes due to 1) local geometric ambiguity and 2) small fracture surfaces. We discuss the two challenges and the limitations of our approach below:

• *Local geometric ambiguity* (left four examples) result in the misplacement of fragments such as the red and yellow fragments on a mug handle in the first column. The third column presents an exceptionally difficult scenario with 20 fragments. As the number of fragments increases, the likelihood of encountering fragments with similar local geometries also rises. Nonetheless, our method uses global shape priors to assemble all fragments into a coherent global shape.

• *Small fracture surfaces* (right two examples) make it hard to assess the fitness of fragments. In the right examples, the bases of two objects are successfully reassembled but fail to connect to the main bodies due to the thin structures serving as connections. Specifically, the right-most column illustrates a 180-degree rotation error in the base of the wine glass.

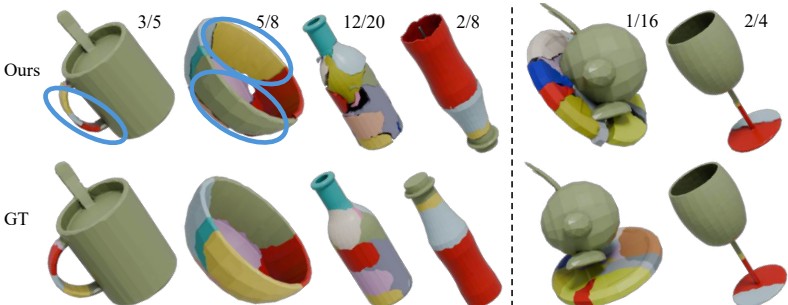

Figure 7: Two popular failure modes are local geometric ambiguity (left four examples) and small fracture surfaces (right two examples).

## 5 CONCLUSION

This paper introduces PuzzleFusion++, an advanced framework for 3D fracture assembly. Our key contributions are a fully neural auto-agglomerative design that simulates human cognitive strategies for puzzle solving and a diffusion model enhanced with feature embedding designs that directly estimate 6-DoF alignment parameters. Extensive quantitative and qualitative experiments show that PuzzleFusion++ outperforms existing methods with significant margins on the Breaking Bad dataset. Future work will focus on increasing inference speed and scaling the approach to more complex assembly tasks involving up to 100 fragments.

ACKNOWLEDGMENTS

This research is partially supported by NSERC Discovery Grants, NSERC Alliance Grants, and John R. Evans Leaders Fund (JELF). We thank the Digital Research Alliance of Canada and BC DRI Group for providing computational resources.

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

# Appendix: PuzzleFusion++: Auto-agglomerative 3D Fracture Assembly by Denoise and Verify

The appendix provides the remaining system details and additional experimental results. Please also refer to our project page `https://puzzlefusion-plusplus.github.io` for a detailed video demonstration of PuzzleFusion++'s auto-agglomerative assembly process.

## A  ADDITIONAL IMPLEMENTATION DETAILS

This section presents the remaining implementation details that are omitted in the main paper.

### A.1  DETAILS OF THE FRAGMENT AUTOENCODER

Figure 8 shows the reconstruction results of our autoencoder. We employ PointNet++ for self-supervised pre-training of the autoencoder, using a Single-Scale Grouping PointNet++ encoder $E$ to aggregate features. Each fragment is represented by a point cloud with 1,000 uniformly sampled points. The point cloud is normalized so the center of mass is at the origin and the longest dimension of the axis-aligned bounding box becomes a unit length. The point cloud is initially encoded into 25 distinct point latent vectors representing local features with dimension 64. Each point latent $z_p$ is associated with the local center coordinates $c_p$.

We apply vector quantization (VQ) for regularization. We use a codebook containing 1024 embeddings, each with 16 dimensions. The encoder outputs point latents mapped to 4 codebook entries. Then, these 4 codes, each with 16 dimensions reshaped back to 64 dimensions, become the regularized local point embedding $z_p$.

The decoder $D$ takes the point latent $z_p$ and its corresponding local center coordinates $c_p$. Then, the MLP layer reconstructs each local latent vector to a local point cloud. We then shift all local reconstructed point clouds based on the corresponding local center positions. The reconstructed points are the union of all local reconstructions:

$$P^{\text{rec}} \leftarrow \{D(z_p) + c_p \mid p\}.$$

We follow the training objectives of VQ-VAE, while using bidirectional Chamfer Distance for the reconstruction loss between the original fragment point cloud $P$ and the reconstructed $P^{\text{rec}}$:

$$\text{Loss} = \text{CD}(P_p, P_p^{\text{rec}}) + \|\text{sg}[E(c_p)] - z_p\|_2^2 + \beta\|E(c_p) - \text{sg}[z_p]\|_2^2, \forall(p)$$

The parameter $\beta$ is set to $0.25$. After training the autoencoder, we keep only the encoder and codebook for our SE(3) denoiser to encode the fragment shape. During denoiser training, the weights of the encoder and codebook are frozen.

### A.2  DETAILS OF THE NOISE SCHEDULER

We have presented our noise scheduler in Figure 3 of the main paper. Similar to humans solving jigsaw puzzles, accurately aligning local fracture surfaces is usually more challenging than knowing the rough location of a fragment. Therefore, we allocate more denoising budgets to getting precise alignments than moving fragments to the rough locations by designing the piecewise function. Figure 9 plots the noise scheduling curves for the linear scheduler, the cosine scheduler, and our customized scheduler. Figure 10 provides the visualization of the denoising process for a vase model. Figure 11 and Table 8 provide the qualitative and quantitative comparisons of the three noise schedulers, respectively.

Our noise scheduler is used for both training and testing, as we follow the vanilla DDPM formulation rather than EDM (Karras et al., 2022). With our tailored noise scheduler for the 3D shape assembly task, the denoising process allocates more steps to refining precise local adjustments rather than finding the rough global location (at test time). As for training, this scheduler indeed makes the process more efficient, as fewer training iterations are spent on "less important" timesteps. If the default (linear or cosine) scheduler were used for training while our scheduler was applied during testing, similar results might still be achieved but would require more training iterations.

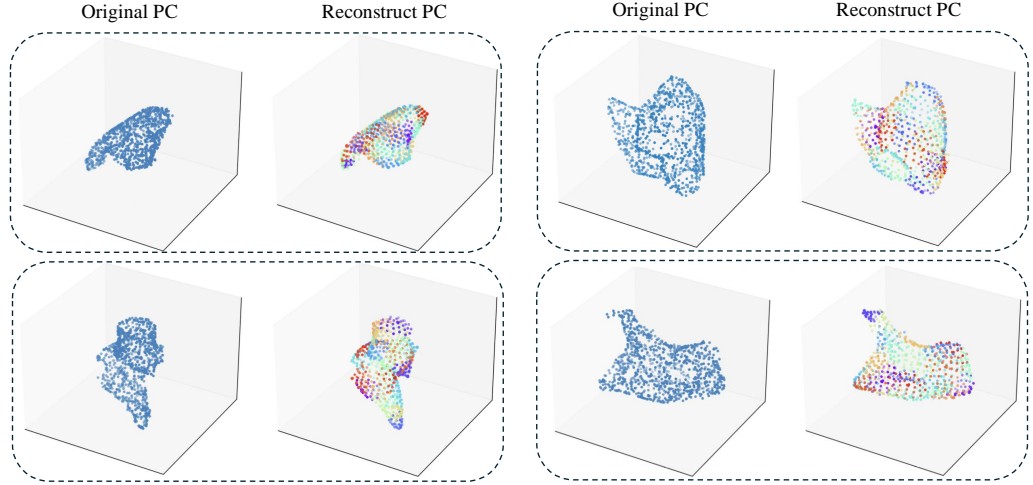

Figure 8: **Reconstruction results of our point clouds VQ-VAE.** The comparison highlights the quality differences between the original point cloud and the reconstructed point cloud generated by the VQ-VAE. In the reconstructed point cloud, the same color represents points reconstructed by the same latent. Each latent is responsible for reconstructing only a local area. This figure demonstrates that the key features of fracture fragments can be preserved in the latent codes.

To illustrate the advantages of our scheduler during the testing stage, we compare it to the linear and cosine scheduler. The linear scheduler uses most of its steps (T=1000 to 150) for rough localization, while the cosine scheduler allocates more denoising steps in the final adjustment phase, outperforming the linear scheduler by a clear margin .

Building on this idea, our scheduler dedicates an even larger portion of the denoising steps to the final adjustment, further improving results. The top three rows of Figure 11 show simpler cases of objects comprising at most 5 fragments. While all the schedulers achieve 100% part accuracy, gaps between fragments are visible for the linear or the cosine schedulers. Our precise alignments may have minimal effects on the standard metrics but significantly enhance the quality of the final assembly.

Table 8: **Comparing different noise schedulers.** Quantitative evaluations of the three noise schedulers shown in Figure 9. Our customized scheduler spends more denoising steps at low noise levels, achieving the best numbers in all the metrics.

| Scheduler | RMSE (Rot.) ↓ | RMSE (Trans.) ↓ | PA ↑ | CD ↓ |
|---|---|---|---|---|
| Linear | 48.2 | 11.9 | 57.5 | 9.21 |
| Cosine | 43.4 | 9.24 | 64.4 | 7.10 |
| Ours | 40.8 | 9.06 | 67.3 | 6.45 |

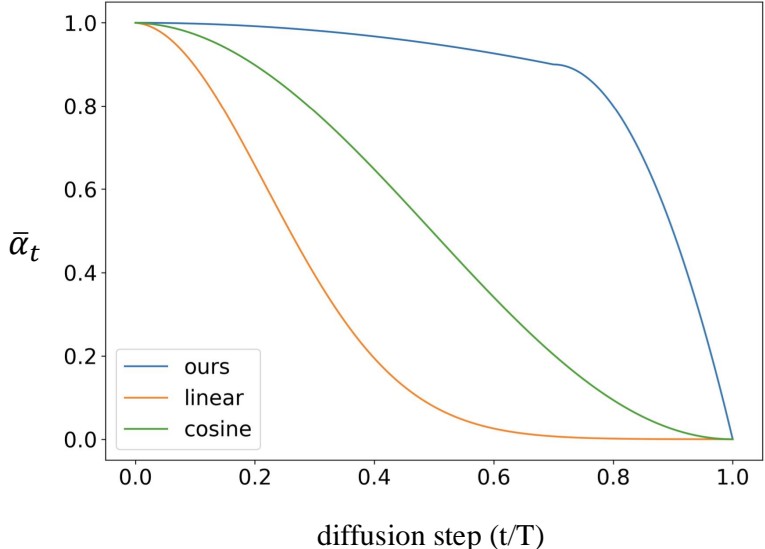

Figure 9: **Comparing different noise schedulers.** Our customized scheduler adds large noise near the end of the diffusion steps. During denoising, we spend more iterations at low noise levels, figuring out precise final alignments.

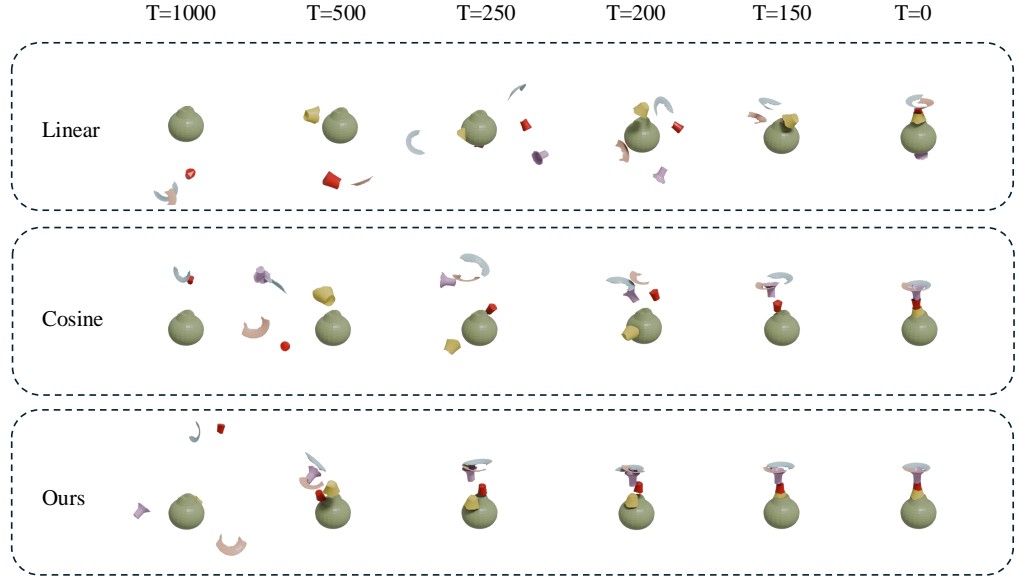

Figure 10: **Comparing different noise schedulers.** The figure shows the denoising process for a vase model for the three noise schedulers shown in Figure 9. The linear scheduler uses most of the denoising steps (T=1000 to T=150) to find a rough location, leaving a few steps for alignment. The cosine scheduler improves efficiency but still wastes steps on rough localization. In contrast, our scheduler quickly identifies the rough location and dedicates more steps to precise alignment.

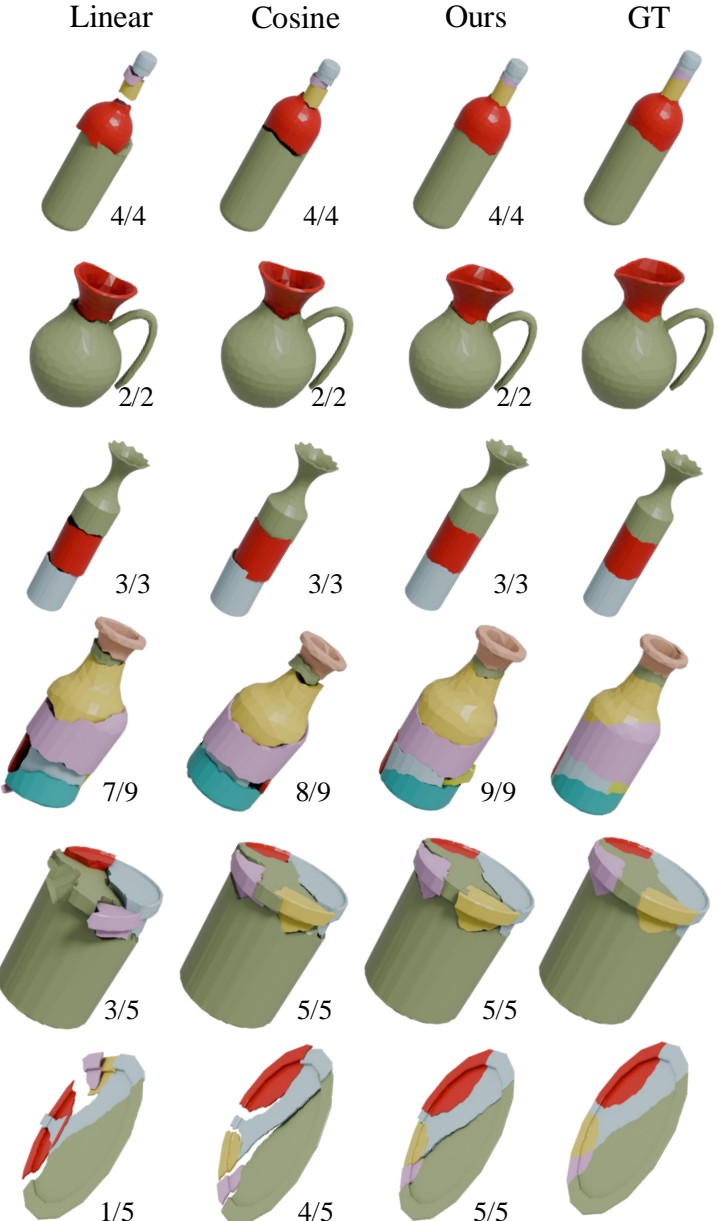

Figure 11: **Comparing different noise schedulers.** The figure shows the final assembly results of the three noise schedulers shown in Figure 9. The first three rows highlight simple cases where our scheduler achieves precise alignment, surpassing others with visible gaps. The last three rows showcase challenging cases, demonstrating the superior performance of our scheduler.

## B  ADDITIONAL EXPERIMENTAL RESULTS AND ANALYSES

### B.1  MORE ANALYSIS ON EVERYDAY OBJECT SUBSET

**More qualitative results.** We provide additional qualitative results based on the number of fragments: Figure 12 for 2-5 fragments, Figure 13 for 6-10 fragments, Figure 14 for 11-15 fragments, and Figure 15 for 16-20 fragments. In the main paper, we have demonstrated that Jigsaw is the primary competing method to our method, so we exclude other baselines to save space.

The additional qualitative results show that Jigsaw yields good results on simple objects with fewer than 6 fragments. When the number of fragments increases, PuzzleFusion++ consistently surpasses Jigsaw. For both approaches, the performance declines as the number of fragments increases, potentially attributed to the local geometric ambiguity discussed in §4.4 of the main paper.

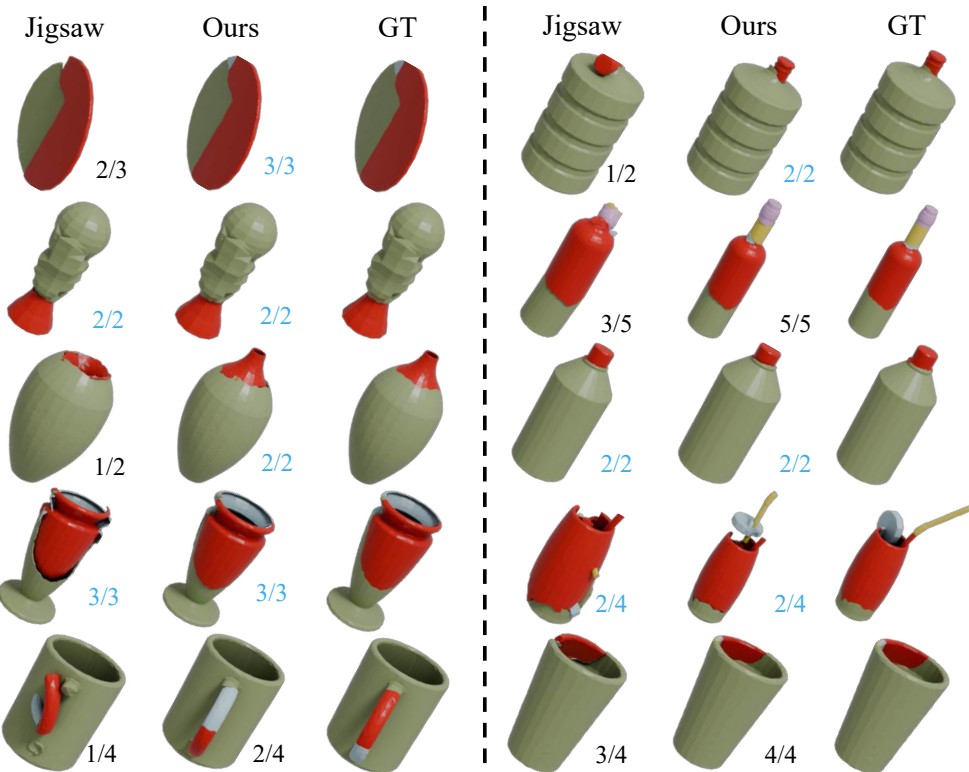

Figure 12: More final assembly results for objects comprising 2 to 5 fragments.

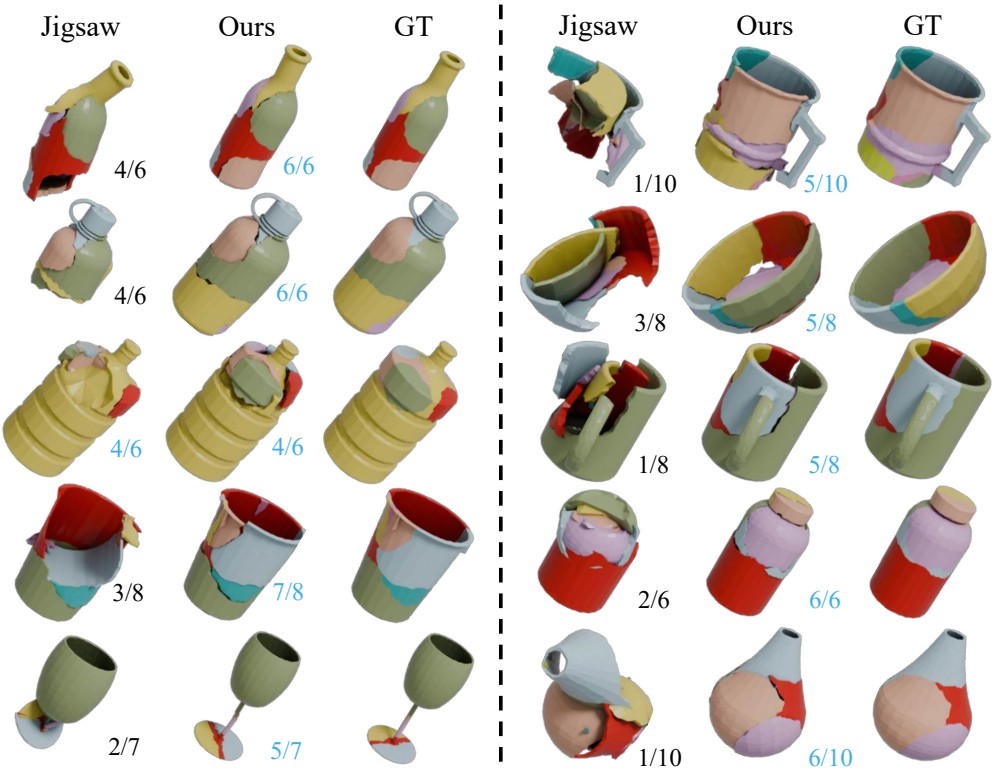

Figure 13: More final assembly results for objects comprising 6 to 10 fragments.

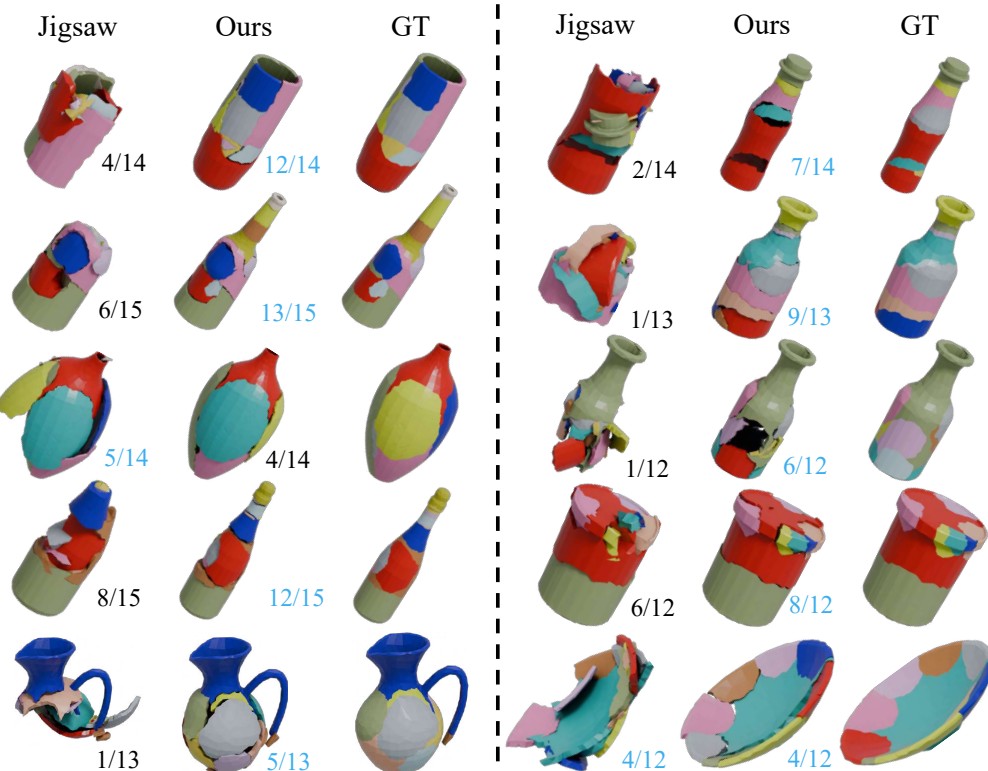

Figure 14: More final assembly results for objects comprising 11 to 15 fragments.

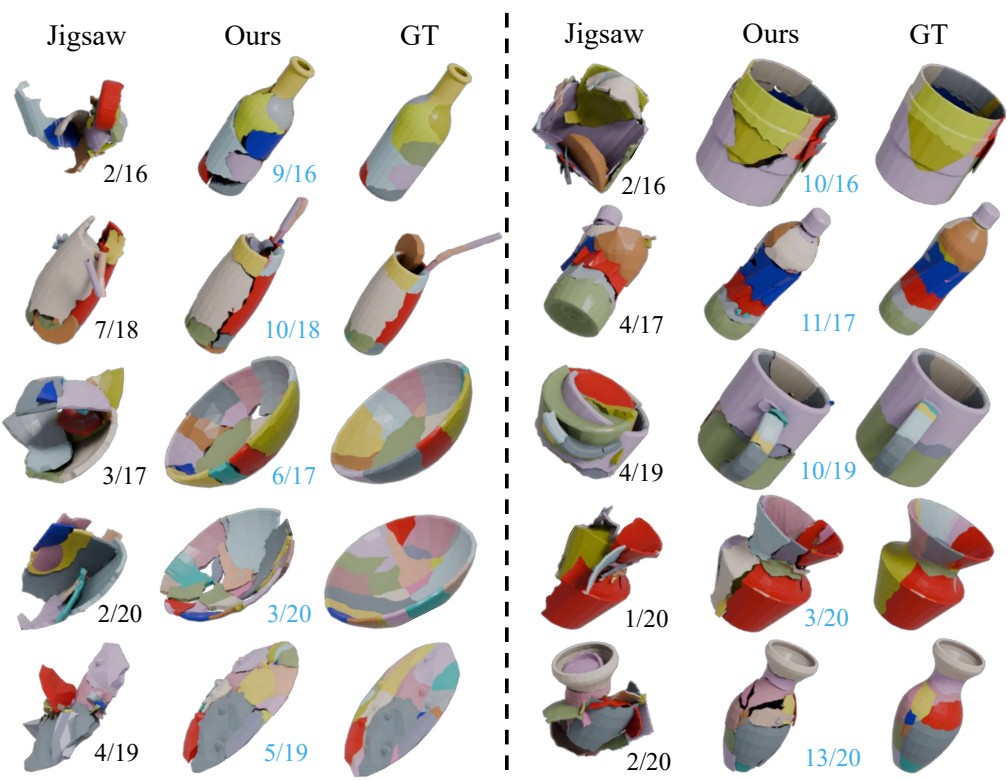

Figure 15: More final assembly results for objects comprising 16 to 20 fragments.

**More quantitative analysis.** Figure 16 shows the distribution of the part accuracy metric with respect to the number of fragments compared with Jigsaw. Our performance drops when the number of fragments increases, as there could be many small pieces and, leading to much higher difficulty. However, compared to Jigsaw, our performance is still significantly better for a large number of fragments.

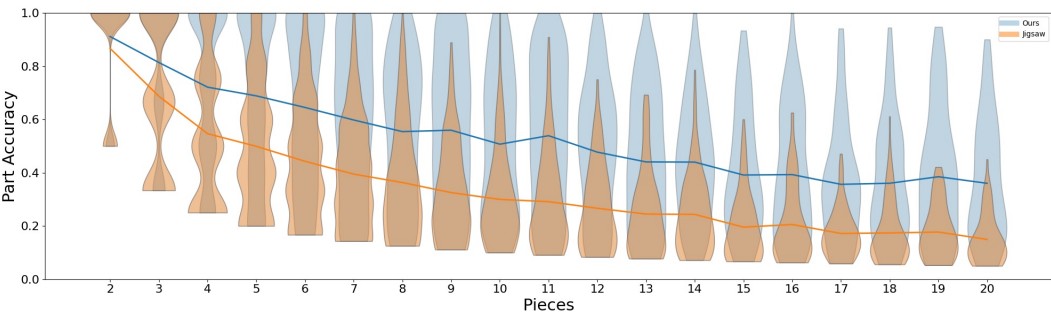

Figure 16: Part accuracy grouped by the number of fragments: Results are tested on the everyday object subset. Shaded areas show part accuracy distribution of our method and Jigsaw. Solid lines indicate the average. Our method presents clear superiority for complex objects with more fragments.

## B.2 MORE ANALYSIS ON COMPLICATED OBJECT

The Everyday subset contains relatively simple objects, such as bottles, glasses and wine. To explore the model's ability to handle more complex shapes, we focused on the Artifact subset, which features challenging archaeological objects with intricate structures. To evaluate this, we fine-tuned the model, initially trained on the Everyday subset, on the Artifact subset.

Table 9 presents the quantitative evaluation on the Artifact subset, comparing our method with baseline approaches: Global, LSTM, and DGL. Our method achieves significantly better performance than these methods. Following this, Figure 17 showcases the qualitative results, further demonstrating the effectiveness of our approach on the challenging object.

Table 9: The Artifact subset of the Breaking Bad dataset contains objects with complicated shapes. We trained the models on the Everyday subset, fine-tuned on the Artifact subset, then evaluated on the Artifact subset in this table.

| Method | RMSE (Rot.) ↓ | RMSE (Trans.) ↓ | PA ↑ | CD ↓ |
|---|---|---|---|---|
| Global | 83.8 | 16.6 | 19.0 | 13.3 |
| LSTM | 84.6 | 16.8 | 21.5 | 11.7 |
| DGL | 81.7 | 16.6 | 17.3 | 19.4 |
| Ours (#$ite$=1) | 41.1 | 9.87 | 62.6 | 9.29 |

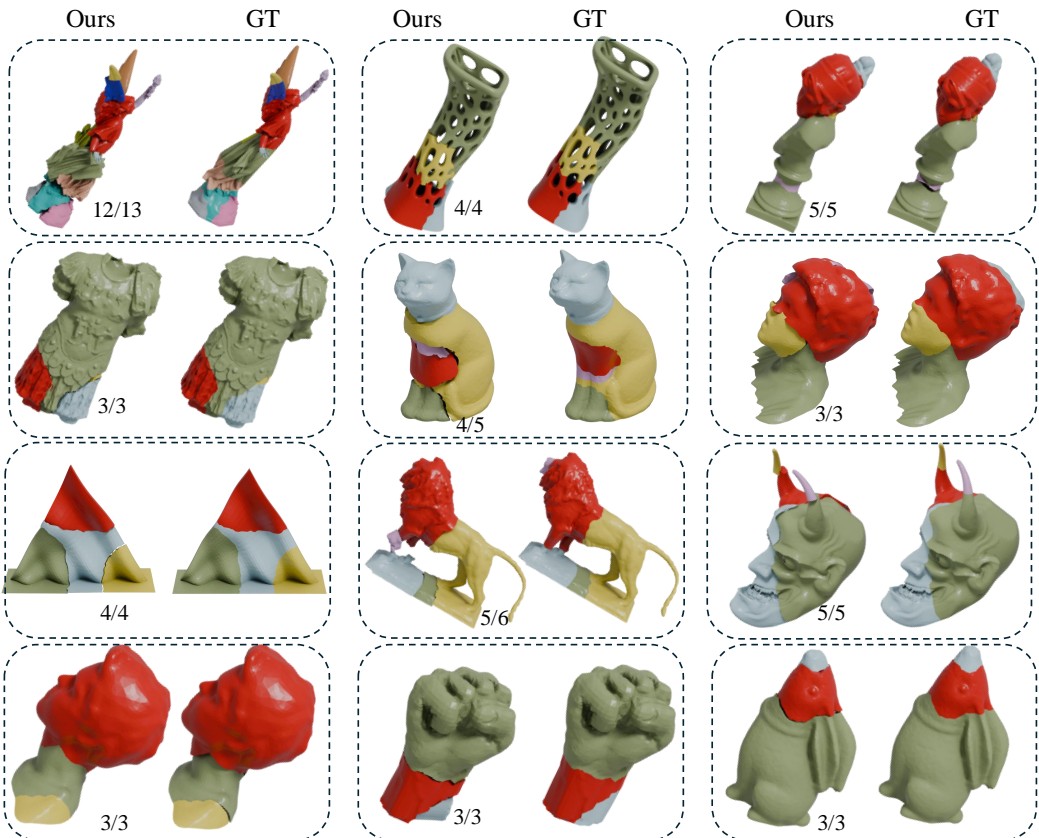

Figure 17: Final assembly results of 12 complex objects from the Artifact subset, including intricate archaeological artifacts such as statues, animal figures, and ornamental designs.

### B.3 COMPARED WITH OTHER RECENT BASELINES

**3D Geometric Shape Assembly via Efficient Point Cloud Matching.** PMTR (Lee et al., 2024) evaluated their method on a simplified version of the data in the Breaking Bad dataset, called the volume-constrained version. This version filters out fragments below a minimum volume threshold, effectively removing very small fragments and making the dataset less challenging. To ensure a fair evaluation, we trained our method on the same volume-constrained Everyday subset and reported the results in Table 10. Our method outperforms PMTR across all metrics, except for the rotation error (RMSE(R)), where it lags by a small margin of approximately 1 degree.

Table 10: PMTR Lee et al. (2024) used a simplified version of the Breaking Bad dataset, called the volume-constrained version. The version filters out fragments below a minimum volume threshold, making the version easier than the standard set. For fair comparison, the table compares our system and PMTR on the volume-constrained version of the Everyday subset.

| Method | RMSE (Rot.) ↓ | RMSE (Trans.) ↓ | PA ↑ | CD ↓ |
|---|---|---|---|---|
| PMTR(Lee et al., 2024) | 31.57 | 9.95 | 70.6 | 5.56 |
| Ours | 32.70 | 5.41 | 78.9 | 3.01 |

**Multi-Fragment Assembly Using Proxy-Level Hybrid Transformer.** PHFormer (Cui et al., 2024) employs a hybrid attention module to model the relationships between fragments. We compared our performance with theirs in the Table 11.

Table 11: Comparison of our method and PHFormer performance on the Everyday subset

| Method | RMSE (Rot.) ↓ | RMSE (Trans.) ↓ | PA ↑ | CD ↓ |
|---|---|---|---|---|
| PHFormer | 26.1 | 9.3 | 50.7 | 9.6 |
| Ours | 38.1 | 8.04 | 70.6 | 6.02 |

Our method outperforms PHFormer in three metrics but has a higher RMSE in rotation.

**Fracture Assembly with Segmentation and Iterative Registration.** FRASIER (Kim et al., 2024) reassembles fractured objects using fracture surface segmentation and iterative registration. It uses GeoTransformer (Qin et al., 2022) to match points between fragment pairs and samples 50k points per fragment. This high point density allows FRASIER to capture detailed fracture surfaces but does not align with the standard 1k-point setup used by baseline methods.

To infer FRASIER's performance under the 1k-point setting, we refer to GeoTransformer's reported performance in Jigsaw (Lu et al., 2023) (Appendix E.3). With 1k points, GeoTransformer produces poor results (RMSE (Rot.) = $84.8°$, RMSE (Trans.) = $14.3 \times 10^{-2}$, PA = 3.1%). Since FRASIER depends on GeoTransformer for registration, it likely cannot deliver reasonable results under the 1k-point-per-fragment setting.

### B.4 ANALYSIS ON HIGH ROTATION ERRORS

Table 6 shows the upper bound of our method, which leverages the ground truth verifier. However, the rotation error remains high (34-degree error). We perform further analysis compared our method and Jigsaw using the results from Table 1. Table 12 shows the improvement margin for rotation is relatively modest (+9.93%), whereas other metrics show significant gains exceeding 20%.

Table 12: Improvement margins (Delta) calculated as the relative performance difference between our method and Jigsaw. Positive values indicate that our method outperforms the baseline.

| | RMSE (Rot.) ↓ | RMSE (Trans.) ↓ | PA ↑ | CD ↓ |
|---|---|---|---|---|
| Delta | +9.93% | +24.86% | +23.21% | +54.66% |

In addition, the failure cases illustrated in Figure 7 are also highly related to the high rotation error:

- **Local geometric ambiguity**: Similar geometry across fragments makes it difficult to determine precise rotations.
- **Small fracture surfaces**: Tiny pieces often lack distinct surface features, leading to 180-degree rotation errors.

All these results demonstrate that our diffusion-based method does not handle rotation as effectively as traditional optimization-based approaches. We provide some thoughts below:

The other three metrics heavily rely on an accurate translation/placement of fragment pieces, where the diffusion-based approach excels by learning global shape priors together with local alignments. On the contrary, the RMSE of rotation mainly examines the accuracy of fine-grained alignments, which relies more on accurate local shape matching. It seems that methods like Jigsaw can produce better rotation-level alignments by conducting direct optimization based on the local surface matching results, while our diffusion-based approach allocates most of the learning capacity for global shapes (correct translations). We believe that a potential direction for improving the rotation-level accuracy can be adding an additional stage that focuses on refining the rotation parameters.

