# OpenReview forum: "PuzzleFusion++: Auto-agglomerative 3D Fracture Assembly by Denoise and Verify"
_ICLR.cc/2025/Conference — ICLR 2025 Poster_

### Official Review · Reviewer_KmDi · 2024-10-27

**Soundness:** 3
**Presentation:** 3
**Contribution:** 3
**Rating:** 8
**Confidence:** 5

**Summary:**

This paper looks at the task of object reassembly in which a set of fractures are given and the goal is to assemble them into an object. This is a challenging problem and has applications in computer vision, computer graphics and robotics. The paper proposes a method called puzzlefusion++ which solves the problem by aligning and merging the fragments into larger groups (think of this as a clustering processing) and then iteratively completing the assembly. Results show that the proposed method make some improvements over the competing methods.

**Strengths:**

1. the proposed method makes sense and I believe this is the way to solve this problem as in doing some clustering first and then merging them (unlike prior work which tries to predict a pose for each fragment in one single pass).

2. performance is good compared to state-of-the-art methods. this means that the proposed method is effective

3. presentation is good and the paper is easy to follow

**Weaknesses:**

1. speed is significantly slower than some methods (see table 1)

2. can the proposed method solve the reassembly task when the number of fractures is big (eg 100)

3. how well can one expect the proposed method to work on unseen objects?

**Questions:**

good paper, but i still have quesitons. please see the weaknesses above

minor question: which software, library is used to render images? the rendering is very professional

---

> ### Author Response · Authors · 2024-11-18
> **Response to Reviewer KmDi**
>
> We thank the reviewer for the construction comments and the questions. We will address the questions and concerns raised by the reviewer below.
>
> **Q1: Speed is significantly slower than some methods**
>
> We thank you for pointing out the speed limitation. We will discuss this question from two perspectives.
>
> - **The use case of shape assembly probably does not require a fast solver.** As discussed in the literature on 3D shape assembly, the main use cases are archeological artifact reconstruction, forensic object reassembly, and protein structure analysis for drug discovery. In these applications, the users care much more about the final assembly quality than the running speed. Jigsaw, the strongest baseline, also needs a rather long running time to solve for one object.
>
> - **Employ more advanced diffusion-based generative models to accelerate the inference.** The relatively slow speed is mainly due to the denoiser requiring multiple denoising steps. Recent advances in diffusion models, flow matching, consistency models, etc. (such as [A], [B], and [C]) indicate that there is still significant room for improving the sampling speed with fewer sampling steps. Our current setup uses 20 denoising steps, suggesting the potential for up to a 20× speed improvement. This is a promising direction for future work, and we will include the discussion in the paper.
>
> **Q2: Can the proposed method solve the reassembly task when the number of fractures is big (eg, 100)**
>
> We followed the same setting as the baselines and focused on ≤20 fractures. While we achieved SOTA performance for 20 fractures, the results are far from perfect. With 100 fractures, local geometric ambiguities would become significantly more severe, amplifying the issues observed in our failure cases section (Sec.4.4) and likely resulting in very poor performance. Therefore, we did not investigate cases with more fracture pieces in our experiments. Handling more fractures is an important direction for future work.
>
> **Q3: How well can one expect the proposed method to work on unseen objects?**
>
> We thank you for the question. We tested our method on the Artifact subset (unseen) of the Breaking Bad dataset, as shown in the main table under "Trained on the everyday subset, tested on the artifact subset." The results are reasonable compared to our main baseline, Jigsaw.
>
> However, PuzzleFusion++ experiences a performance decline across all metrics compared to Jigsaw on the unseen objects. This is because PuzzleFusion++ learns global spatial priors by the diffusion model that simultaneously solves the arrangements of all the pieces. The global priors are effective for everyday objects but struggle to generalize to the different categories in the Artifact subset. In contrast, Jigsaw focuses on local geometry learning, making it less sensitive to major category changes.
>
> To further investigate, we finetuned our denoiser on the Artifact subset and provide qualitative results in Figure 17 and quantitative results in Table 9. With less than 20% of the training iterations, the model can be adapted to previously unseen object categories with good performance. Together with the generalization results in Table 1, we believe our method has reasonable ability to handle unseen objects
>
> **Software/Library for rendered images:**
>
> We render our results using BlenderToolBox[D]. We will release our rendering code together with other parts of the project code.
>
> ***
> [A] Lu, C., & Song, Y. (2024). Simplifying, Stabilizing and Scaling Continuous-Time Consistency Models. *arXiv preprint arXiv:2410.11081*.
>
> [B] Lipman, Y., Chen, R. T., Ben-Hamu, H., Nickel, M., & Le, M. (2022). Flow matching for generative modeling. arXiv preprint arXiv:2210.02747.
>
> [C] Karras, T., Aittala, M., Aila, T., & Laine, S. (2022). Elucidating the design space of diffusion-based generative models. *Advances in neural information processing systems*, *35*, 26565-26577.
>
> [D] Derek Liu. BlenderToolBox. Available at: https://github.com/HTDerekLiu/BlenderToolbox

---

> > ### Comment · Reviewer_KmDi · 2024-11-25
> >
> > My concerns have been addressed. I am enthusiastic about this paper and remain my score as an 8.

---

> ### Author Response · Authors · 2024-11-26
>
> Thank you for your feedback and support of our work! We are glad our responses addressed your concerns.

---

### Official Review · Reviewer_P4T7 · 2024-10-28

**Soundness:** 2
**Presentation:** 3
**Contribution:** 2
**Rating:** 6
**Confidence:** 5

**Summary:**

This paper introduces PuzzleFusion++, which addresses the 3D fracture assembly problem. It employs a diffusion model enhanced by a proposed autoencoder, optimized denoise scheduling, denoiser structure, and an auto-agglomerative verifier. These advancements collectively contribute to performance that surpasses that of previous work.

**Strengths:**

The paper demonstrates performance improvements over previous work. I like the idea of using a verifier to assess the quality of the assembly results.

**Weaknesses:**

Despite significant efforts on ablation studies of the proposed method, the paper does not clearly differentiate from prior works. A valuable contribution would be for the authors to provide (component level) comparisons with similar methods, such as PuzzleFusion and DiffAssemble. It would be beneficial to highlight that what limits the existing approaches and what contributes to the success of the proposed one. For now, the uniqueness of this paper is not clear.

**Questions:**

1. With the name of this paper, it’s hard not to compare it with PuzzleFusion that have a similar motivation in tackling jigsaw puzzle related problem. Could PuzzleFusion be adapted to handle 3D tasks as a baseline for comparison? What distinguishes PuzzleFusion++ from PuzzleFusion, particularly in features that enhance its suitability for 3D tasks?

2. The matching from Jigsaw is not perfect, is there a scheme to handle such situation? Meanwhile, the provide upper bound based on perfect matching is only 34 degree error. This is still very high. If Jigsaw can have such perfect matching, the error can be reduced to less than 10 degrees. This raises questions about the suitability of using a diffusion model for this type of problem.

3. Even for comparison among diffusion models, why PuzzleFusion++ can significantly outperforms DiffAssemble? Can the authors describe the key difference that make your model works better?

---

> ### Author Response · Authors · 2024-11-20
> **Response to Reviewer P4T7**
>
> We thank you for the construction comments and the questions.
>
> **W1 & Q1.1: Could PuzzleFusion be adapted to handle 3D tasks as a baseline for comparison?**
>
> PuzzleFusoin is specifically designed for 2D jigsaw puzzles with simple polygonal shapes, where a 1D chain of 2D coordinates represents each puzzle piece. The approach (including its puzzle piece encoder and the denoising network) cannot be directly used for the 3D shape assembly task, where each piece is a set of unordered 3D points.
>
> **W1 & Q1.2: What distinguishes PuzzleFusion++ from PuzzleFusion, particularly in features that enhance its suitability for 3D tasks?**
>
> The 3D fracture assembly task poses several challenges: i) The 6-DoF solution space is more complicated than the 3-DoF solution space of 2D spatial puzzles; ii) Unlike 2D puzzle solvers that mainly leverage image semantics or polygon structures, 3D fracture assembly requires a deep understanding of fracture surfaces; iii) There can be many small 3D fragments, further complicating the problem.
>
> PuzzleFusion++ proposes new designs to address the above challenges. Specifically, our VQ-VAE with PointNet++ can encode fine details of fracture surfaces into a set of local latents, boosting the shape understanding of the denoising network; Our auto-agglomerative framework verifies assembly results and composes confident small pieces into larger ones to facilitate future iterations, which effectively improves the assembly success rate; We also design tailored components for the diffusion model, including the noise scheduler and the denoising transformer with local shape encodings as condition.
>
> **Q2.1: The matching from Jigsaw is not perfect, is there a scheme to handle such situation?**
>
> As shown in the Table 6, the verifier's performance is not optimal. We cannot address most of the errors made in Jigsaw point matching. However, our verifier leverages a Transformer to incorporate all pairwise matching information between fragments. This design enables the model to reason globally across multiple pairs, potentially overcoming local matching errors produced during Jigsaw matching.
>
> **Q2.2: Upper bound rotation error is still very high (34 degree error)**
>
> Thank you for this great observation. We agree on this point — the high rotation error with even the GT matching indeed indicates the limitation of the diffusion-based approach for super-accurate local alignments.
>
> Following your observation, we do more analysis using the results from Table 1. As presented in the table below, the improvement margin for rotation is relatively modest (+9.93%), whereas other metrics show significant gains exceeding 20%.
>
> |  | RMSE (Rot.) ↓ | RMSE (Trans.) ↓ | PA ↑ | CD ↓ |
> | --- | --- | --- | --- | --- |
> | Delta | +9.93% | +24.86% | +23.21% | +54.66% |
>
> In addition, the failure cases illustrated in **Figure 7** are also highly related to the high rotation error:
>
> 1. **Local geometric ambiguity**: Similar geometry across fragments makes it difficult to determine precise rotations.
> 2. **Small Fracture Surfaces**: Tiny pieces often lack distinct surface features, leading to 180-degree rotation errors.
>
> All these results demonstrate that our diffusion-based method does not handle rotation as effectively as traditional optimization-based approaches. We provide some thoughts below:
>
> The other three metrics heavily rely on an accurate translation/placement of fragment pieces, where the diffusion-based approach excels by learning global shape priors together with local alignments. On the contrary, the RMSE of rotation mainly examines the accuracy of fine-grained alignments, which relies more on accurate local shape matching. It seems that methods like Jigsaw can produce better rotation-level alignments by conducting direct optimization based on the local surface matching results, while our diffusion-based approach allocates most of the learning capacity for global shapes (correct translations).
>
> We believe that a potential direction for improving the rotation-level accuracy can be adding an additional stage that focuses on refining the rotation parameters. We will include this discussion in the paper.
>
> **W1 & Q3: Why does PuzzleFusion++ significantly outperform DiffAssemble, even among diffusion models? Can the authors describe the key difference that make your model works better?**
>
> We presents our core design of denoiser in the ablation section (L454-L466), which investigates 3 core modules of our SE3 denoiser. 1) We have the pre-traiend PointNet++ VQ-VAE to encode local geometric information, while DiffAssemble only encodes a global semantic latent for each fragment, thus lacking fine-grained local details. 2) DiffAssemble does not take care of the ambiguity of 3D rigid transformation, while we use the anchor fragment to avoid the training ambiguity. 3) We have a tailored noise scheduler to handle the 3D fracture assembly task while DiffAssemble simply keeps the default designs of DDPM.

---

> > ### Comment · Reviewer_P4T7 · 2024-11-22
> >
> > I appreciate the authors' response to my questions. After reading the rebuttal and other reviewers' comments, I will keep my rating. I also hope authors can add the discussion on the upperbound rotation error to the paper.

---

> ### Author Response · Authors · 2024-11-25
> **Response to Reviewer P4T7**
>
> We appreciate your feedback and suggestion. We have included the rotation discussion in Appendix B.4 in the updated PDF version.

---

### Official Review · Reviewer_6kWy · 2024-10-31

**Soundness:** 3
**Presentation:** 3
**Contribution:** 3
**Rating:** 6
**Confidence:** 4

**Summary:**

This paper proposes PuzzleFusion++, a framework for 3D fracture assembly. A fully neural auto-agglomerative design is proposed that simulates human cognitive strategies for puzzle solving. Moreover, a diffusion model enhanced with feature embedding designs is devised that directly estimate 6-DoF alignment parameters.

**Strengths:**

1.	This paper is well-written.
2.	The motivation is clear enough.
3.	The organization of this paper is great.
4.	The alignment verifier is well designed.

**Weaknesses:**

1.	The Figure 3 is a little puzzled. A better format is recommended.
2.	The showed 3D objects seems relatively simple. More complicate objects from Objaverse are commended.

**Questions:**

1.	The main difference between PuzzleFusion and PuzzleFusion++ should be clarified in the paper, since it is a future work from PuzzleFusion.
2.	Are 25 points enough to represent a fragment in Sec.3.1?
3.	Does the pairwise alignment verifier work well for all objects?
4.	In auto-agglomerative inference, are 6 iterations enough for merging? Have you try more iterations?
5.	Please discuss could reinforcement learning whether is help for this task.
6.	As mentioned in Weakness, could you provide more complicate objects during rebuttal?
7.	Please discuss the possible solution for your limitations.

---

> ### Author Response · Authors · 2024-11-20
> **Response to Reviewer 6kWy (1/3)**
>
> We thank you for the construction comments and the questions.
>
> **W1: The Figure 3 is a little puzzled. A better format is recommended.**
>
> We thank for the suggestion. Reviewer ox6f also mentioned making Figure 3 more concise. We have updated pdf with new version Figure 3. We reformat figure 3 to the inference pipeline instead of seperate components.
>
> **W2&Q6: As mentioned in Weakness, could you provide more complicate objects during rebuttal?**
>
> We thank for the suggestion. Artifact is the other subset in the Breaking Bad dataset which contains many complicated objects than the Everyday subset. To have a proper evaluation on the Artifact subset, we take the model pretrained on the Everyday subset, and finetune it on the training split of the Artifact subset with only 20% of the pretraining iteration, and then evaluate the model on the test split of Artifact subset.
>
> The qualitative and quantitative results are provided in **Figure 17** and **Table 9**, showcasing the model's performance on these more challenging objects
>
> **Q1: The main difference between PuzzleFusion and PuzzleFusion++ should be clarified in the paper, since it is a future work from PuzzleFusion.**
>
> PuzzleFusion is the first work that employs diffusion models to solve 2D spatial puzzles, demonstrating the potentials of diffusion models as general iterative solvers for non-generation or discriminative tasks. Our paper further extends diffusion models to more complicated 3D fracture assembly problem.
>
> At architecture level, PuzzleFusion is specifically designed for 2D jigsaw puzzles with simple **polygonal shapes**, where a 1D chain of 2D coordinates represents each puzzle piece. The approach cannot be used for the 3D shape assembly task, where each piece is a set of unordered 3D points.
>
> In addition, the 3D fracture assembly task poses several challenges: i) The 6-DoF solution space is more complicated than the 3-DoF solution space of 2D spatial puzzles; ii) Unlike 2D puzzle solvers that mainly leverage image semantics or polygon structures, 3D fracture assembly requires a deep understanding of fracture surfaces; iii) There can be many small 3D fragments, further complicating the problem. Our PuzzleFusion++ proposes new designs to address these challenges. More specifically, our VQ-VAE with PointNet++ can encode fine details of fracture surfaces into a set of local latents, boosting the shape understanding of the denoising network; Our auto-agglomerative framework verifies assembly results and composes confident small pieces into larger ones to facilitate future iterations, which effectively improves the assembly success rate; We also design tailored components for the diffusion model, including the noise scheduler and the denoising transformer with local shape encodings as condition.

---

> ### Author Response · Authors · 2024-11-20
> **Response to Reviewer 6kWy (2/3)**
>
> **Q2: Are 25 points enough to represent a fragment in Sec.3.1?**
>
> Following the experimental setting in the literature, each fragment piece has 1000 sampled points, and our PointNet++ VQ-VAE encode the 1000 points into the 25-point latent vector. Each point here has a latent embedding that captures local shape details. One way to better understand whether the 25-point latent representation is to examine the quality of the reconstructed point cloud. As shown in Figure 8, the visualization demonstrates that the decoded point clouds retain the overall shape and essential details of the original fragments.
>
> **Q3: Does the pairwise alignment verifier work well for all objects?**
>
> We have included the pairwise alignment verifier's performance ablation in Table 6. The verifier does not work well for all objects. To handle this, we only classified scores > 0.9 as the true label, achieving 90.77% accuracy, 87.88% precision, and 45.95% recall. This high threshold ensures that the selected pairs have high confidence, lowering the possibility of wrong merging.
>
> Additionally, Table 5 shows the upper-bound performance of our method using the ground truth verifier. The ground truth verifier performs significantly better than the verifier using Jigsaw matchings, highlighting the potential for further improving the verifier.
>
> **Q4: In auto-agglomerative inference, are 6 iterations enough for merging? Have you try more iterations?**
>
> Yes, the 6 iterations are enough for merging. We tested with more iterations and found that increasing the iterations did not improve performance.
>
> Below are the results of more auto-agglomerative iterations:
>
> | Iterations | RMSE (Rot.) ↓ | RMSE (Trans.) ↓ | PA ↑ | CD ↓ |
> | --- | --- | --- | --- | --- |
> | 6 | 38.1 | 8.04 | 70.6 | 6.02 |
> | 7 | 38.5 | 7.94 | 70.3 | 6.48 |
> | 8 | 38.9 | 8.06 | 69.9 | 6.70 |
> | 9 | 38.8 | 8.01 | 70.1 | 6.60 |
> | 10 | 38.7 | 7.94 | 70.2 | 6.35 |

---

> ### Author Response · Authors · 2024-11-20
> **Response to Reviewer 6kWy (3/3)**
>
> **Q5: Please discuss could reinforcement learning whether is help for this task.**
>
> We believe RL is less suitable for this task due to the challenges of defining an effective reward function. The final reward can only be provided after completing all assembly steps, making it hard to optimize the learning process. This limitation could slow down learning and reduce the overall system performance. We will include this discussion in our final version.
>
> **Q7: Please discuss the possible solution for your limitations.**
>
> **Local Geometric Ambiguity**: The issue of local geometric ambiguity arises because some fracture surfaces are too similar and “confuse” the model. A possible solution is to increase the number of sampled 3D points for each fracture (currently we use 1000, the default setting inherited from previous works in the literature). By increasing the point cloud density, the PointNet++ encoder can capture more accurate local geometric details, and the denoising network can have better chance to distinguish these similar fracture surfaces.
>
> **Small Fracture Surfaces**: Small fracture surfaces can result in failed connections between merged fragments. A potential solution is to enable the network to learn more global information. If the network can imagine the shape of original object, it can correctly place the fragments in their intended positions.
>
> We will include this discussion into the future work.

---

> ### Comment · Reviewer_6kWy · 2024-11-25
> **Response to Authors**
>
> Thank you for the clarifications and for addressing my concerns. I will keep my rating as positive.

---

> > ### Author Response · Authors · 2024-11-26
> >
> > We thank you for your valuable feedback and appreciate your time in reviewing our paper. We are glad to have addressed your concerns.

---

### Official Review · Reviewer_XQS8 · 2024-10-31

**Soundness:** 2
**Presentation:** 3
**Contribution:** 2
**Rating:** 6
**Confidence:** 5

**Summary:**

Authors improve over existing work on 3d fracture assembly by proposing an iterative approach. The proposed method iteratively denoised poses for current fragments and merge aligned fragments into clusters until all fragments are merged into a single 3D object. They build over existing work (Scarpellini et. al, Wu et al.) regarding the diffusion model formulation for alignment. The novelty regards adopting a transformer for clustering fragments.

**Strengths:**

- Authors' proposal is easy to follow and well-thought
- Noise scheduler improves performances by a surprising margin. Since different noise schedulers do not change training dynamics and effects only MC estimate of the loss (Kingma et al. 2024), my guess it's that this scheduler makes training more efficient. I'd like to see more comments on that, either in main paper or supplementary. Right now it's relegated to a small section in supplementary and not discussed in main paper.
- Good ablations section
- Method is somewhat novel. There's no technical novelty since it adopts existing methods and approaches (iterative refinement has been adopted by Ken et al, diffusion model formulation was in Scarpellini et al). Still the combination of these modules make outperform the proposed baselines so I would not consider this a weakness

Kim, Jinhyeok, Inha Lee, and Kyungdon Joo. "Fracture Assembly with Segmentation And Iterative Registration." ICASSP 2024-2024 IEEE International Conference on Acoustics, Speech and Signal Processing (ICASSP). IEEE, 2024.

Kingma, Diederik, and Ruiqi Gao. "Understanding diffusion objectives as the elbo with simple data augmentation." Advances in Neural Information Processing Systems 36 (2024).

**Weaknesses:**

- L158  anchor fragments were introduced in Jigsaw (Lu eta al) and are not adopted in original BreakingBad. Comparing anchor-based methods to methods that do not adopt anchors is a bit unfair. A fair comparison would entail re-training those baselines with anchors (e.g., L212-214 regarding alignment could be adopted in all other baselines). I believe this point makes the experimental section weaker and does not reflect the actual performances of the baselines.
- Missing baselines: authors should also compare to other existing methods that were published before ICLR deadline: PHFormer: Multi-Fragment Assembly Using Proxy-Level Hybrid Transformer (Cui et al, AAAI2024), Fracture Assembly with Segmentation And Iterative Registration (Kim et al., ICASSP2024). Both achieve impressive results and are not cited by the authors--especially the latter achieves RMSE R 23.8, RMSE T 7.30, PA 74.50.

**Questions:**

- L150-L154 what guarantees that the input representation are equivariant (aka "sensitive to rotation"). Normalizing wrt center mass achieves invariance, but not necessarily equivariance.

---

> ### Author Response · Authors · 2024-11-21
> **Response to Reviewer XQS8 (1/2)**
>
> We appreciate the reviewer’s constructive and detailed feedback. We try to address the concerns and questions below.
>
> **Strengths 2: More discussion/analysis on the noise scheduler.**
>
> We thank you for the insightful comment. We agree that our analysis of the scheduler was limited. We have added quantitative and qualitative comparisons of three schedulers (i.e., linear, cosine, and ours) together with the visualizations of the denoising process in **Table 8** and **Figure 11** and **Figure 10**.
>
> Our noise scheduler is used for both training and testing, as we follow the vanilla DDPM formulation rather than EDM[A]. With our tailored noise scheduler for the 3D shape assembly task, the denoising process allocates more steps to refining precise local adjustments rather than finding the rough global location (at test time). As for training, this scheduler indeed makes the process more efficient, as fewer training iterations are spent on "less important" timesteps. If the default (linear or cosine) scheduler were used for training while our scheduler was applied during testing, similar results might still be achieved but would require more training iterations.
>
> To illustrate the advantages of our scheduler during the testing stage, we compare it to the linear and cosine scheduler. The linear scheduler uses most of its steps (T=1000 to 150) for rough localization (Figure 10), while the cosine scheduler allocates more denoising steps in the final adjustment phase, outperforming the linear scheduler by a clear margin (Table 8).
>
> Building on this idea, our scheduler dedicates an even larger portion of the denoising steps to the final adjustment, further improving results (Table 8). The top three rows of Figure 11 show simpler cases of objects comprising at most 5 fragments. While all the schedulers achieve 100% part accuracy, gaps between fragments are visible for the linear or the cosine schedulers. Our precise alignments may have minimal effects on the standard metrics but significantly enhance the quality of the final assembly.
>
> **W1: Fair comparison regarding anchor fragments**
>
> We believe the reviewer's main concern is that the post-processing step introduced in Jigsaw could lead to unfair evaluations. This step involves using the largest fragment to align the predicted assembly with the ground truth before metric calculation. Both our method and Jigsaw employ this post-processing. We also noticed that the recent ICML'24 paper "3D Geometric Shape Assembly via Efficient Point Cloud Matching", as mentioned by Reviewer ox6f, also uses this setting. This is the "proper" test setting -- those early baseline methods should have done it correctly.
>
> To correct the testing setting of those baselines, we applied the same post-processing step to all of them (See the table below). It is evident that our method still significantly outperforms the baselines. We also included the numbers reported in the original submission in brackets.
>
> | Method | RMSE(R)↓ | RMSE(T)↓ | PA↑ | CD↓ |
> | --- | --- | --- | --- | --- |
> | Global | 62.02 (**80.7**) | 19.43 (**15.1**) | 30.35 (**24.6**) | 18.9 (**14.6**) |
> | LSTM | 62.34 (**84.2**) | 21.41 (**16.2**) | 28.32 (**22.7**) | 23.4 (**15.8**) |
> | DGL | 61.91 (**79.4**) | 19.21 (**15.0**) | 33.50 (**31.0**) | 15.0 (**14.3**) |
> | SE3 | 61.03 (**79.3**) | 19.04 (**16.9**) | 28.13 (**8.41**) | - |
> | Ours | 38.10 | 8.04 | 70.60 | 6.03 |
>
> Regarding whether a method uses the anchor fragment during training, we think it's the design choice of each method and has nothing to do with fairness. We added this design component to handle the ambiguity of rigid 3D transformation for our method, and we are not responsible for adding this to all baselines. Fairness should be defined by whether the task input and output are the same across methods. Our input consists of fragments’ point clouds with random poses, and the output is the final alignment parameters for each point cloud, which are identical to those of the baselines. We do not require additional information about the anchor fragment or its ground truth pose beforehand.

---

> ### Author Response · Authors · 2024-11-21
> **Response to Reviewer XQS8 (2/2)**
>
> **W2: Missing baselines**
>
> We apologize for the oversight. We checked the two papers and discussed them here and will include them in our final version.
>
> - **PHFormer**:
>
>     The AAAI paper introduces PHFormer, which employs a hybrid attention module to model the relationships between fragments. We compared our performance with theirs in the table below:
>
>     | Method | RMSE (Rot.) ↓ | RMSE (Trans.) ↓ | PA ↑ | CD ↓ |
>     | --- | --- | --- | --- | --- |
>     | PHFormer | 26.1 | 9.3 | 50.7 | 9.6 |
>     | Ours | 38.1 | 8.04 | 70.6 | 6.02 |
>
>     Our method outperforms PHFormer in three metrics but has a higher RMSE in rotation. We will also include these quantitative and additional qualitative comparisons with PHFormer in the final version.
>
> - **Fracture Assembly with Segmentation And Iterative Registration**:
>
>     The ICASSP paper introduces FRASIER, a framework for reassembling fractured objects using fracture surface segmentation and iterative registration. For the registration stage, FRASIER uses the GeoTransformer[B] to match points between fragment pairs.
>
>     Unlike prior methods that use 1k points per fragment, FRASIER samples 50k points per fragment. This high point density enables detailed fracture surface capture but does not align with the standard 1k-point setup used by other methods. Moreover, since their code is not available, we cannot evaluate their performance under the 1k-point setting.
>
>     To infer FRASIER’s performance with 1k points per fragment, we refer to GeoTransformer’s reported performance in Jigsaw (Appendix E.3). With 1k points, GeoTransformer produces poor results (RMSE (Rot.) = 84.8°, RMSE (Trans.) = 14.3 × 10⁻², PA = 3.1%). These findings show that GeoTransformer is ineffective under the 1k-point setting. Since FRASIER relies on GeoTransformer for registration, it likely cannot achieve reasonable results under the 1k-point-per-fragment setting.
>
>
> **Q1: What guarantees that the input representation are equivariant?**
>
> The input representation is equivariant because of the way we pre-trained our VQ-VAE. During pretraining, we applied random rotations to the input point cloud and used the rotated point cloud for supervision. This ensures that the latent embedding reflects the rotation applied to the input, achieving equivariance empirically.
>
> ---
>
> [A] Karras, T., Aittala, M., Aila, T., & Laine, S. (2022). Elucidating the design space of diffusion-based generative models. *Advances in neural information processing systems*, *35*, 26565-26577.
>
> [B] Qin, Z., Yu, H., Wang, C., Guo, Y., Peng, Y., Ilic, S., ... & Xu, K. (2023). Geotransformer: Fast and robust point cloud registration with geometric transformer. *IEEE Transactions on Pattern Analysis and Machine Intelligence*, *45*(8), 9806-9821.

---

> > ### Comment · Reviewer_XQS8 · 2024-11-25
> >
> > Thanks for answering all my concerns and providing a details response. I hope the authors did not misunderstand my comments on fairness. I understand that it's not responsibility of these authors the errors of past papers, but I believe that research should be crystal clear on why previous methods failed, and there was no mention to the different protocols. I'm still skeptical that applying anchor-alignment post-training to the baselines would reflect their actual performances on the anchor-aligned protocol. Still, I appreciate authors' commitment to improve the manuscript and provide experiments and I raise my score

---

> > > ### Author Response · Authors · 2024-11-26
> > >
> > > We are delighted that our rebuttal addressed your concerns. We thank you for your valuable feedback and for raising your score. We will highlight the protocol differences (anchor vs. non-anchor) in the final version.

---

### Official Review · Reviewer_ox6f · 2024-11-04

**Soundness:** 3
**Presentation:** 3
**Contribution:** 3
**Rating:** 8
**Confidence:** 4

**Summary:**

This paper proposes PuzzleFusion++, an "auto-agglomerative" method for the task of 3D fracture assembly.
Specifically, the proposed pipeline undergoes a iterative process of using a diffusion model to predict a 6-DoF alignment for each fragment, followed by a transformer model which verifies and merges pairwise alignments into larger ones.
This is much alike how humans assemble fragments - hypothesizing how two fragments fit together, and checking if it the alignment is indeed ture.
To encode fragments into latent vectors suitable for diffusion training, the authors integrate PointNet++ and VQVAE.
PuzzleFusion++ achieves state of the art on the Breaking Bad dataset by a large margin, and comprehensive analysis clarifies the significance of each introduced module.

**Strengths:**

- Novel approach ideated as 'auto-agglomerative', which has been implemented through the integration of a diffusion model (SE3 denoiser) and a transformer model (pairwise alignment verifier).

- The manuscript is overall well-written and easy to follow.

- Comprehensive analysis experiments, which validates the design choices of PuzzleFusion++ and clarifies how each design choice affects the performance.

- Strong performance on the Breaking Bad benchmark, outperforming existing methods by a large margin.

**Weaknesses:**

- While Figure 3 provides valuable details into how the SE3-denoiser and Pairwise Alignment Verifier work, it is not very visual, and not straightforward to understand as-is. I believe providing Figure 3 as a form of an algorithm may improve its readability.

- The authors mention that "different anchor initialization does not affect the quality of the assembly results (L482)"; however, it can be seen that the results in Table 7 is closer to the results for Ours(#ite=1) in Table 2. I believe it is an overstatement to mention that anchor initialization 'does not affect' the quality of the assembly results - unless a single iteration (#iteration=1) was used for the results in Table 7. It would be informative to provide the results for varying number of iterations for random initializations of the anchor fragment.

- The paper is missing a recent baseline for 3D assembly, which seem to show strong performances on the Breaking Bad benchmark as well:
Nahyuk Lee et al, 3D Geometric Shape Assembly via Efficient Point Cloud Matching, ICML 2024.


- Minor writing mistakes:
1) Table 4: Autoncoder -> Autoencoder
2) L 454: Denioser -> Denoiser

**Questions:**

- Why was the number of iterations set to 6? Table 2 shows that the results are best at # iterations = 6. At what number of iteration does PuzzleFusion++ achieve the best results, without further increase in performance with increasing number of iterations?

- In Table 4, what are the results when the scheduler is x (i.e., not the proposed scheduler), while the Autoencoder and Anchor fragment are being used? It would be helpful to include this result in the ablation, for improved clarity of the ablation results.

- Figure 9 visualizes the difference between 3 noise schedulers - could the authors include a quantitative/qualitative comparison of all 3 noise schedulers? The proposed rationale of "locating more denoising budgets to getting precise alignments than moving fragments to the rough locations" sounds intuitive, and it would certainly help to have additional results to validate this claim, beyond the results in Table 4.

The idea of the proposed PuzzleFusion++ is interesting and novel, and also shows strong empirical results. While the paper can be still improved by including more recent baselines and providing more comprehensive results, I believe that the strengths of the paper outweigh its weaknesses as-is. I am leaning towards accept, and am willing to improve my score if my questions / weaknesses are addressed.

---

> ### Author Response · Authors · 2024-11-18
> **Response to Reviewer ox6f**
>
> We thank the reviewer for the constructive and detailed feedback. We provide the response to each point below.
>
> **W1: Providing Figure 3 as a form of an algorithm may improve its readability.**
>
> We thank for the suggestion. We are working on redesigning Figure 3, and we first address the remaining questions/concerns in this thread. We will update Figure 3 in the pdf shortly and send a follow-up message.
>
> **W2: Overstatement to mention that anchor initialization 'does not affect' the quality of the assembly results**
>
> We apologize for the confusion — Table 7 only shows results with #iteration=1, so it actually supports our claim about the robustness over different anchor initializations. To clarify, we add results for #iteration=2 and #iteration=4 with random anchor initializations. Similar to the setup of Table 7, we run the inference 10 times and calculate the mean and variance. For better readability, we also included the numbers reported in Table 2 of the main paper in brackets. This confirms the quality consistency across different initializations.
>
> |  | RMSE (Rot.) ↓ | RMSE (Trans.) ↓ | PA ↑ | CD ↓ |
> | --- | --- | --- | --- | --- |
> | #ite=1 | 40.86 ± 0.37 **(40.8)** | 9.03 ± 0.18 **(9.06)**   | 67.49 ± 0.51 **(67.3)** | 6.69 ± 0.62 **(6.45)** |
> | #ite=2 | 39.30 ± 0.26 **(39.4)**    | 8.53 ± 0.06 **(8.48)** | 68.9 ± 0.18 **(68.8)** | 6.29 ± 0.16 **(6.28)** |
> | #ite=4 | 38.51 ± 0.15 **(39.1)**    | 8.21 ± 0.13 **(8.23)** | 70.0 ± 0.20 **(69.8)** | 6.17 ± 0.12 **(6.15)** |
>
> **W3: Missing baseline**
>
> We thank you for pointing out the ICML paper. We apologize for the oversight.
>
> The ICML paper presents a new method, PMTR, which employs an efficient high-order feature transformation layer to establish reliable correspondences. When investigating the paper and the official implementation, PMTR used an easier "volume-constrained" subset of the Breaking Bad dataset, where fragments below a minimum volume threshold are excluded. Most previous works, including our submission, use the original, more difficult dataset. To make a fair comparison, we train and evaluate our method using the volume-constrained subset and provide the results below:
>
> | Method | RMSE(R)↓ | RMSE(T)↓ | PA↑ | CD↓ |
> | --- | --- | --- | --- | --- |
> | PMTR | 31.57 | 9.95 | 70.6 | 5.56 |
> | Ours (6 iterations) | 32.70 | 5.41 | 78.9 | 3.01 |
>
> PuzzleFusion++ outperforms PMTR on all metrics except rotation error (RMSE(R)) by a small margin (approximately 1 degree). Please refer to Appendix B.3 for more details.
>
> **W4: Typos**
>
> We thank you for pointing out these. We fixed the typos in the updated PDF.
>
> **Q1: Optimal performance of the number of iterations.**
>
> The maximum iteration was set to 6 as it is close to convergence, with minimal or no consistent improvements further. Below are the results for more iterations.
>
> | Iterations | RMSE (Rot.) ↓ | RMSE (Trans.) ↓ | PA ↑ | CD ↓ |
> | --- | --- | --- | --- | --- |
> | 6 | 38.1 | 8.04 | 70.6 | 6.02 |
> | 7 | 38.5 | 7.94 | 70.3 | 6.48 |
> | 8 | 38.9 | 8.06 | 69.9 | 6.70 |
> | 9 | 38.8 | 8.01 | 70.1 | 6.60 |
> | 10 | 38.7 | 7.94 | 70.2 | 6.35 |
>
> These results confirm that iteration 6 achieves nearly optimal performance, with further iterations yielding negligible or inconsistent changes.
>
> **Q2: What are the results when the scheduler is x**
>
> The ablation of the noise scheduler had already been included in the first row of Table 4 in the initial submission. When the scheduler is "×," we use a linear scheduler by default.
>
> **Q3: Could the authors include a quantitative/qualitative comparison of all 3 noise schedulers?**
>
> We appreciate the suggestion. We updated the PDF by adding a quantitative comparison of the linear, the cosine, and our schedulers in **Table 8** and a qualitative comparison in **Figure 11**.
>
> Additionally, we include visualizations of the denoising process with different schedulers in **Figure 10**. These visualizations further illustrate how the linear and cosine schedulers spend more time finding the rough locations of fracture pieces, while ours focus more on precise alignment.

---

> > ### Author Response · Authors · 2024-11-23
> > **Reformat Figure 3**
> >
> > Thank you for your patience. We have updated Figure 3 in the PDF, redesigning it as an inference pipeline instead of separate components for denoiser and verifier. We hope this revised version improves readability and aligns better with your suggestion.

---

> ### Comment · Reviewer_ox6f · 2024-11-25
>
> Thank you for your detailed response! I believe my concerns and questions have been addressed adequately. I have raised my recommendation.

---

> > ### Author Response · Authors · 2024-11-26
> >
> > Thank you so much for your valuable feedback! We are pleased that our rebuttal addressed your concerns and sincerely appreciate the time and effort you put into reviewing our paper.

---

### Author Response · Authors · 2024-11-23
**General Response**

We sincerely appreciate the reviewers' time and effort in providing detailed and insightful feedback on our submission. The revisions to our paper are summarized in five main aspects.

1. **Improve the Readability of Figure 3 (ox6f, 6kWy):** We reformat Figure 3 as an inference pipeline rather than separate architectural details for the denoiser and verifier. We hope this version improves readability.
2. **Provide More Complicated Objects Results (6kWy, KmDi):** We include qualitative and quantitative results on more complex objects in Figure 17 and Table 9. We provide more training and evaluation details in Appendix B.2.
3. **Detailed Analysis on Noise Scheduler (ox6f, XQS8):** We provide detailed analysis of the noise scheduler in Appendix A.2. We add quantitative and qualitative comparisons of three schedulers (i.e., linear, cosine, and ours) along with visualizations of the denoising process in Table 8, Figure 11, and Figure 10.
4. **Missing Baselines (ox6f, XQS8):** We include discussions and comparisons with recent baselines in Appendix B.3.
5. **Rotation Error Analysis (P4T7):** Our method shows higher rotation error with a ground truth verifier. We provide more analysis in Appendix B.4.

---

### Meta-Review · Area_Chair_4AGg · 2024-12-23

**Metareview:**

This paper proposes a 3D fracture assembly method that learns a diffusion model to predict a 6-DoF alignment for each fragment iteratively, followed by a transformer model that verifies and merges pairwise alignments into larger ones. Unlike previous diffusion-based models, it simulates how humans assemble fragments gradually and check the validity of the alignment. To encode fragments into latent vectors suitable for diffusion training, it integrates PointNet++ and VQVAE. The proposed method achieves state-of-the-art on the Breaking Bad dataset by a large margin, and comprehensive analysis clarifies the significance of each introduced module.
All reviewers appreciated the novelty of the proposed auto-agglomerative approach and its comprehensive analyses. The main concerns raised by reviewers were unclear exposition, unfair experimental setups, and missing comparisons. The authors’ detailed rebuttal addressed most of them, resulting in unanimous acceptance at the end of the discussion. AC thus recommends acceptance.

**Additional Comments On Reviewer Discussion:**

The main concerns raised by reviewers were unclear exposition, unfair experimental setups, and missing comparisons. The authors’ detailed rebuttal addressed most of them such that all reviewers either remained positive or raised their scores after discussion.

---

### Decision · Program_Chairs · 2025-01-22

Accept (Poster)